# Interpersonal and Intimate Violence in Mexican Youth: Drug Use, Depression, Anxiety, and Stress during the COVID-19 Pandemic

**DOI:** 10.3390/ijerph20156484

**Published:** 2023-07-31

**Authors:** Silvia Morales Chainé, Gonzalo Bacigalupe, Rebeca Robles García, Alejandra López Montoya, Violeta Félix Romero, Mireya Atzala Imaz Gispert

**Affiliations:** 1Psychology Faculty, National Autonomous University of Mexico, Mexico City 04510, Mexico; alelopez.unam@gmail.com (A.L.M.); violetaflix@gmail.com (V.F.R.); 2Department of Counseling and School Psychology, College of Education and Human Development, University of Massachusetts Boston, Boston, MA 02125, USA; gonzalo.bacigalupe@umb.edu; 3National Institute of Psychiatry “Ramón de la Fuente Muñiz”, Ministry of Health, Mexico City 14370, Mexico; reberobles@hotmail.com; 4General Directorate of Community Attention, National Autonomous University of México, Mexico City 04510, Mexico; mimaz@unam.mx

**Keywords:** victimizing and perpetrating violence, interpersonal and intimate violence, harmful alcohol and drug use, mental health symptoms, paths of violence, gender

## Abstract

The COVID-19 pandemic may have increased interpersonal and intimate violence, harmful use of alcohol and other drugs (AODs), and mental health problems. This study uses a valid path model to describe relationships between these conditions of young Mexicans during the second year of the pandemic. A sample of 7420 Mexicans ages 18 to 24—two-thirds of whom are women—completed the Life Events Checklist, the Alcohol, Smoking, and Substance Involvement Screening Test, the Major Depressive Episode Checklist, the Generalized Anxiety Scale, and the Post-traumatic Stress Disorder (PTSD) Checklist. Young Mexicans reported higher rates of victimization and perpetration of interpersonal and intimate violence and mental health symptomatology than those noted pre- and in the first year of the pandemic. The harmful use of AOD rates were similar to those reported by adolescents before. The findings suggest asymmetric victimization and perpetration of intimate violence by gender (with women at a higher risk). More men than women have engaged in the harmful use of AODs (except for sedatives, which more women abuse). More women than men were at risk of all mental health conditions. The path model indicates that being a victim of intimate violence predicts the harmful use of tobacco, alcohol, cocaine, and sedatives, depression, anxiety, and specific PTSD symptoms (such as re-experimentation and avoidance symptoms). Being a victim of interpersonal violence resulted in severe PTSD symptoms (including avoidance, negative alterations in cognition-mood, and hyperarousal signs). The harmful use of sedatives predicted depressive symptoms. Men’s victimizing intimate violence model contrasted with that of women, which included being the victim of interpersonal violence and severe PTSD symptoms. The high school youth model had three paths: victimizing intimate violence, victimizing interpersonal abuse, and sedative use, which predicted depression. Our findings could serve as the basis for future studies exploring the mechanisms that predict violence to develop cost-effective preventive programs and public policies and to address mental health conditions during community emergencies.

## 1. Introduction

The COVID-19 pandemic may have been accompanied by an increase in interpersonal violence, harmful use of alcohol and other drugs (AODs), and mental health problems [1,2]. However, reports on the prevalence or incidence of violence, AODs, and mental health conditions have been based on separate studies conducted at the start of the pandemic or found in data obtained earlier, suggesting unclear directionality on relationships between these harmful effects. Clarity on the relationship between involvement in interpersonal and intimate violence, AODs, and suffering from mental health problems in youth will make it possible to design effective, efficient public health policies and preventive intervention strategies during health emergencies.

### 1.1. Violence, AOD Use, and Mental Health Conditions before and in the First Year of the Pandemic

The Pan-American Health Organization (PAHO) [3] identified higher rates of Years Lived with Disability (YLD) due to interpersonal and intimate violence in America between 2000 and 2019. It observed a rate of 59.8 YLD per 100,000 population, 79.8 in women and 41.2 in men in 2019. According to The World Drug Report of the United Nations Office on Drugs and Crime (UNODC) [4], there was a 26% increase in the prevalence of drugs used between 2010 and 2020, based on previous and initial data during the pandemic. Layman et al. [5], however, found that AOD trends seemed to vary in each country due to the pandemic between 2020 and 2022. Regarding mental health issues, the World Health Organization (WHO) [1] reported a 25% increase in depression and anxiety, while Bourmistrova et al. [6] observed a prevalence of 20.39% for depression, 18.85% for anxiety, and 18.99% for post-traumatic stress disorder (PTSD) symptoms, after one or more months of being severely ill from COVID-19, based on a systematic review of 2019–2021 papers.

In Mexico, PAHO [3] identified higher rates of YLD due to interpersonal and intimate violence between 2000 and 2019. It observed a rate of 64.73 per 100,000 Mexicans, 86.49 for women, and 45.65 for men in 2019. Data from the National Institute of Statistics and Geography (INEGI), [7] identified a four-point increase in total lifetime violence against women between 2016 and 2021 (from 66.1% to 70.10%). In their Mental Health and Substance Abuse Observer System (NCA-MHSAMOS), the National Committee on Addictions [8] noted that, during the pandemic, 2.6% of Mexicans reported experiencing an increase in interpersonal violence, with 19.80%, 18.7%, and 3.1% increasing their use of alcohol, tobacco, and other drugs, respectively. It also concluded that these increases were due to anxiety (15.9%) or stress conditions (17.7%). Reports suggest rising trends and a high prevalence of violence, AODs, and mental health illness. Systematic reviews have suggested relationship directionalities that could be considered to improve public mental health policies and cost-effective preventions and treatment interventions [9]. 

### 1.2. Interpersonal and Intimate Violence

Interpersonal and intimate violence consists of these behaviors within a relationship, causing physical, sexual, or psychological harm and including acts of physical aggression, sexual coercion, psychological abuse, and controlling behaviors [2,10,11,12]. According to Johnson [13], Holtzworth-Munroe and Stuart [14], Weathers, Blake et al. [15], and Scott-Storey et al. [16], interpersonal and intimate violence definitions include being the victim of experiencing and perpetrating physical assault (such as being attacked, hitting, slapping, kicking, beating up, threatening, isolating, or intimately abusing), assault with a weapon (such as being shot, stabbed, threatened or threatening with a knife, gun, or bomb), sexual assault (being raped or raping, attempting rape, or performing any type of sexual act through force or threat of harm), and any other unwanted or uncomfortable sexual experience. Victimizing or perpetrating interpersonal and intimate violence therefore includes everything from the least severe form of violence, through sexual or psychological abuse, to severe mixed violence, both inside and outside the home, exhibiting traits of victimization, or the perpetration of abuse. 

Research has focused on victimizing intimate abuse against women as the most prevalent form of interpersonal violence during the COVID-19 pandemic [11,16]. White et al. [9] systematically reviewed 2012–2020 research and reported higher lifetime intimate violence prevalence rates among women over sixteen than during the previous year. Nearly four out of every ten women reported experiencing intimate violence during their lifetime, and one in four had reported it in the previous year. They concluded that women in the community had the highest prevalence of victimization through physical, psychological, and sexual violence in the previous year compared to clinical groups. Kourti et al. [11] recently reported that the pandemic had caused an increase in domestic violence cases, particularly during the first week of lockdown in 2020. 

Glowacz et al. [17] also studied the types of intimate violence associated with participants’ sex or age during the first year of the pandemic. They reported that the prevalence of victimizing physical assault was higher in men (12.30%), whereas the prevalence of victimizing psychological violence was higher in adult women (35.20%). Scott-Storey and collaborators [16] concluded that it is more important to address forms of violence than the different prevalence between the sexes. They suggested that forms of violence are the result of perceived violence in men and women. They noted that essential differences in how men perceive victimizing intimate violence appear to be more related to emotional and sexual forms than to physical abuse received when, for example, retaliation or marital conflicts involve children as witnesses in conflicts. Glowacz et al. [17] have also posited that younger adults involved in a relationship were more likely to experience or perpetrate physical and psychological violence during lockdown. The authors also showed that younger adults involved in relationships reported anxiety and depression symptoms associated with violence. 

### 1.3. Relationships and Directionality between Violence, AOD Use, and Mental Health Conditions

Studies on the prevalence of substance use and mental health among youth populations during the first year of the pandemic have observed an increase in alcohol, cannabis, non-prescription medical drugs, and nicotine use [18] and high rates of depression, anxiety, and post-traumatic stress disorder (PTSD) among 12- to 18-year-old participants [19]. One in five adolescents, regardless of sex, engaged in regular (once a week or more) use of at least one psychoactive substance, while 52% of adolescents met the clinical criterion for depression, 39% for anxiety, and 46% for PTSD during 2020 [19]. 

The directionality of the association between violence, AODs, and mental health symptoms has been suggested by pre-pandemic data and studies addressing one or two variables in 2020. Brabete and collaborators [20] reported that AOD use, and mental health symptoms are consequences of victimizing intimate violence. Machisa and Shamu [21] pointed out that one in two women who experienced intimate physical or sexual violence had consumed alcohol and that one in four had binge-drunk during the previous year. In Mexico, Morales et al. [22] reported experiences of emotional and physical abuse related to stress, sadness, and anxiety. Victimizing intimate violence has also been associated with an increased likelihood of using marijuana, stimulants, and other psychoactive substance [20]. Craig et al. [19], Glowacz et al. [17], and White et al. [9] also stated that youth experiencing violence at home suffered depression, anxiety, and post-traumatic stress symptoms, before or during the first year of the pandemic. Harmful AOD use and mental health symptoms have therefore been associated with victimizing intimate violence among young men and women, pre-pandemic and in the first year of the pandemic. 

Drug use has also been studied as a predictor of interpersonal and intimate violence and mental health conditions based on pre-pandemic data. AOD use could predict being a victim or perpetrating intimate violence [23,24]. Dos-Santos and collaborators [25] reported that a history of AOD use was associated with being a victim of psychological, physical, and intimate partner sexual violence. Barchi et al. [26] reported that young women were 10.98 times more likely to experience physical intimate violence and 4.6 times more likely to experience psychological violence when both partners had drunk alcohol. In regard to perpetrating violence, Zhong et al. [27] reported a higher odds ratio of violence among those with AOD disorders, based on a systematic 1990–2019 review. The authors observed that individuals with a diagnosed AOD disorder have a 4-to-10-fold higher risk of perpetrating interpersonal violence compared with general populations without a drug use disorder. Cannabis, hallucinogens, stimulants, opioids, and sedatives were associated with a high risk of violence. It seems, however, that interpersonal violence rather than intimate partner violence was the result of AOD use. The magnitude of perpetrated interpersonal violence appears to vary depending on the type of drug used. Being a victim of or perpetrating intimate violence has been attributed to drug use, resulting in poor mental health [28]. In Mexico, binge drinking was related to stress, sadness, and anxiety during the first year of the pandemic [22].

### 1.4. Factors, the Directionality of Associations, and the Social Determinants Yet to Be Explored

Several pre-pandemic reviews have also suggested that socio-demographic conditions make youths more vulnerable to violence, harmful AOD use, and mental health conditions. Being a woman of a certain age or having a certain degree of educational attainment appears to increase the number of episodes of these conditions [29,30,31]. Dos-Santos et al. [25] noted that less than eight years of education was associated with victimizing psychological, physical, and sexual intimate violence.

The association between forms of violence, AOD use, and mental health symptoms has been described with several populations in different directions considering pre-first-year pandemic data. However, reports on the prevalence or incidence of violence, AODs, and mental health conditions have been based on separate studies conducted at the start of the pandemic or found in data obtained earlier without knowing what happened afterwards. Such studies have also suggested an unclear directionality of relationships between these harmful effects. Thus, describing the last prevalence and an integrated model of directionality regarding relationships would help to slow the progression of social determinants, drug abuse, and mental health conditions. The focus on victimizing and perpetrating interpersonal and intimate violence, harmful AOD use, and mental health symptoms during the second year of the pandemic is essential since these conditions can become worst and more significant during emergencies. It is needed to describe their association and design effective public policies and preventive interventions. Several factors, such as being a victim or a perpetrator, the directionality of the associations, and social determinants (such as sex and educational attainment), could shed light on the role of each factor in these links. The validation of the concepts within a predictive model is also essential every time it is necessary to understand a pandemic [16,32,33]. 

The Structural Equation Model (SEM) through confirmed factor analysis (CFA) with its chi-squared and fit indices, based on a fit function given a specific estimation method, is the recommended tool for assessing the validity of relationships between variables [34,35,36]. The indices of a model with a good fit must be under 0.08 for the Root-Mean-Squared Error of Approximation (RMSEA), under 0.06 for the Standardized Root-Mean-Squared Residual (SRMR), and over 0.90 for the Tucker–Lewis Index (TLI) and a Comparative Fit Index (CFI), from the chi-squared test of the SEM. The RMSEA and the SRMR are absolute indices determining the distance between a hypothesized and a perfect model. CFI and TLI are incremental fit indices that compare the fit of the hypothesized model with that of a baseline model (a model with the worst fit). Evaluating relationships with a statistically advanced strategy could shed light on the association between violence, harmful AOD use, and mental health symptoms among Mexican youths, making it possible to design cost-effective community policies during emergency situations based on empirical data. 

The present study uses a valid path model to describe the association between victimizing and perpetrating interpersonal and intimate violence, harmful AODs, and mental health conditions in Mexican youths during the second year of the COVID-19 pandemic. We have several hypotheses (Ha), the main one being that victimizing interpersonal and intimate violence are associated with harmful AOD use and mental health symptoms, moderated by age and education demographics. The study therefore aims to explore whether victimizing interpersonal and intimate violence predicts harmful psychoactive substance use (Ha1), depression (Ha2), anxiety (Ha3), and PTSD symptoms (Ha4), with differences between sex and educational attainment in the context of the pandemic. In addition, it explores whether harmful AODs affects the perpetration of interpersonal and intimate violence (Ha5), depression (Ha6), anxiety (Ha7), and PTSD symptomatology (Ha8). 

## 2. Methods

### 2.1. Participants

Participants were invited to enroll in the web-based application between 1 September 2021 and 31 August 2022, through conferences in the media to obtain a link available on the public Mexican Health Ministry website and the official institutional website of the leading public university in Mexico. Participants were asked to log into the system with their email to identify their participation. Inclusion criteria were being of legal age, having at least completed high school, and residing in Mexico. Exclusion criteria were being under 17 or over 25, not having completed high school, or being a healthcare provider. We also considered the criteria for internet E-surveys, such as data protection, development, testing, contact mode, advertising the survey, mandatory/voluntary participation, completion rate, cookies used, IP check, log file analysis, registration, and atypical timestamp considerations [37]. Since the technological system does not allow non-response rates, 100% of the participants were volunteers who completed the questionnaire. Our sample was therefore not homogeneous. We surveyed 7420 young Mexicans with a mean age of 20 (*SD* = 1.90, range = 18–24), 5106 (68.80%) of whom were women and 2314 (31.20%) men. A total of 1689 (22.80%) reported completing senior high school, while 5731 (77.20%) obtained university degrees (the age averages and standard deviations were the same for both educational attainment levels). The distribution of the total sample by comparison of the variables is shown in Table 1.

We adhered to the privacy policies established in the General Protection of Personal Information in the Possession of Regulated Entities Act [38]. Participants therefore agreed to answer the mental health screening while keeping their data asymmetrically encrypted in the database held in the official university domain. We used security locks to protect the participants’ information and guarantee its management as they agreed.

Regarding informed consent, the researchers informed the participants that the purpose of the survey was to understand mental health risks and how to deal with them. We also told participants that confidentiality would be ensured since we calculated general averages. Participants were told that their participation was voluntary, that findings would be used for epidemiological research, and that they could refuse to comply with data requests and drop out at any point in the study. Although incentives were not offered, immediate written feedback was provided through a programmed algorithm, including psychoeducational tools (such as infographics, videos, and Moodle^®^ courses on COVID-19, self-care, relaxation techniques, problem-solving, and socio-emotional management skills). Phone numbers were provided to obtain remote psychological counseling by Zoom or phone through the Zoiper^®^ 3.5 switcher from the Health Ministry and public university services. Finally, the benefits of accessing the platforms or requesting help for dealing with mental health conditions were described. A data section, in which participants could provide their phone numbers or emails so they could be contacted, was included to enable them to request remote psychological care. The protocol was approved with the code FPSI/422/CEIP/157/2020 by the Institutional Review Board of the Psychology Faculty Ethics Committee on Applied Research at the National Autonomous University of Mexico.

### 2.2. Instruments

A web-based application ([33], see misalud.unam.mx) included two dichotomic answer–questions on sex and educational attainment (man–woman; high school or university degree) and five psychological tests, using all scales.

The Life Events Checklist 5th Edition (LEC-5), [15,39] included fourteen selected yes/no dichotomic response items on violence from the Post-traumatic Checklist (PCL-5, A criterion [40]) divided into four scales/factors. Four items asked about victimizing interpersonal violence, four about victimizing intimate violence, four about perpetrating interpersonal violence, and two about perpetrating intimate violence, in the previous six months (see Appendix A). The reliability values of the subscales fluctuated between α = 0.68 and α = 0.76 (see Table 2). The confirmatory factor analysis (CFA) found a good test factor structure (e.g., $X^{2}$ [13] = 343.566, *p* < 0.001, an *RMSEA* = 0.059, a *CFI* = 0.949, and a *TLI* = 0.917). Note that each item’s prompt included the origin of the violence (such as physical assault (… being attacked, hit, slapped, kicked, beaten up)) and the origin of the intimate violence (such as was this physical abuse inflicted by a family member or your partner?). We therefore screened for victimizing, perpetrating, interpersonal, or violence by adding a score in each scale. A score over 1 in the violence scales was considered to have met the criterion for victimizing, perpetrating, interpersonal, or intimate violence. If subjects checked a violent event, they were asked to select the one that bothered them most at the time and to answer the questions in part B of the PCL-5 (see below).

The WHO Alcohol, Smoking, and Substance Involvement Screening Test (ASSIST) determines harmful use for ten groups/scales of AODs: tobacco (cigarettes, chewing tobacco, and cigars), alcoholic beverages (beer, wine, and spirits), cannabis (marijuana, pot, grass, and hash), cocaine (coke and crack), amphetamine-type stimulants (speed, meth, and ecstasy), inhalants (nitrous, glue, petrol, paint, and thinner), sedatives or sleeping pills (diazepam, alprazolam, flunitrazepam, and midazolam), hallucinogens (LSD, acid, mushrooms, trips, and ketamine), opioids (heroin, morphine, methadone, buprenorphine, and codeine), and other drugs [41]. ASSIST has proved to have good validity and reliability coefficients. The reliability values fluctuated between α = 0.80 for the alcohol dimension and α = 0.91 for stimulants ([42]; see Table 2). Confirmatory factor analysis (CFA) found a good test factor structure ($X^{2}$ [1583] = 50,863.65, *p* < 0.001, an *RMSEA* = 0.040, an *SRMR* = 0.032, a *CFI* = 0.920, and a *TLI* = 0.913). ASSIST consists of eight questions screening for harmful AOD use, including: (1) lifetime use; (2) use in the past three months; (3) having a strong desire to use the drug in question; (4) health, social, legal, or financial problems; (5) failing to do what is expected because of the use of the drug in question; (6) other expressions of concern about the use of the drug in question; (7) attempts to reduce use of the drug in question; and (8) injecting any drug (non-medical use only). The first item has dichotomous options: yes [1] or no [0]. Items two to five have a five-option response: never; once or twice; monthly; weekly; and daily or almost daily. The value of each response option varies from 0, 2, 3, 4, 6 for item two; 0, 3, 4, 5, 6 for item three; 0, 4, 5, 6, 7 for item four; to 0, 5, 6, 7, 8 for item five. Items six to eight have a three-option response: no, never [0]; yes, it happened in the past three months [6]; or yes, but not in the past three months [3]. The score for harmful use of each substance is calculated by adding the answers to questions two to seven. Neither question five on tobacco nor questions one and eight for all substances were used to calculate the score. A score over four indicates harmful risk criteria for all drugs except alcohol, where the score would have to be over 11 to meet the criteria. Participants reporting injecting drugs were referred to specialized emergency care. 

The Major Depressive Episode (MDE) checklist consists of one scale of eleven five-option-response items ([40], such as Do you feel worthless or not good enough?; see Appendix A). The MDE has good validity and reliability coefficients [43]. The α = 0.92 and the confirmatory factor analysis (CFA) was found to have a good checklist factor structure ($X^{2}$ [32] = 2643.99, *p* < 0.001, an *RMSEA* = 0.067, an *SRMR* = 0.023, a *CFI* = 0.975, and a *TLI* = 0.965). The response options involved how often participants experienced symptoms in the past twelve months: always [1], nearly always [2], sometimes [3], rarely [4], or never [5]. We considered several steps to calculate the total score: part 1, part 2, part 3, and criterion A and B guidelines. The criteria for Part 1 were met when items one and two (Sadness or depressed mood? and Discouraged because of how things are going in your life?) were answered with options 1 or 2. The criteria for Part 2 were met when five or more items were answered with options 1 or 2 from items 2 to 10 plus part 1. The criteria for Part 3 were met when question 3 (Loss of interest or pleasure?) was recorded with response options 1 or 2. Criterion A was met when part 1 and part 2 or 3 were completed. Criterion B was met when question 11 (Symptoms causing impairment in social, occupational, or other important areas of functioning?) was recorded with response options 1, 2, or 3. Finally, an MDE was identified when criteria A and B were met [40]. 

The Generalized Anxiety (GA) scale consists of five eleven-option-response items (adapted from Goldberg and collaborators [44]; such as I have felt nervous or on edge; see Appendix A). The GA has shown good validity and reliability coefficients [43]. The α = 0.94 and the confirmatory factor analysis (CFA) found a good scale factor structure ($X^{2}$ [5] = 350.57, *p* < 0.001, an *RMSEA* = 0.061, an *SRMR* = 0.007, an *CFI* = 0.996, and a *TLI* = 0.992). Response options ranged from zero (total absence of symptom) to ten (full presence of symptoms) for whether participants felt anxious in the past two weeks. We therefore screened for GA by adding the score and dividing it by five. In keeping with Goldberg et al. [44], an average of 60% was considered to have met the criterion for GA.

The PCL-5 consists of twenty-five-option-response items to assess post-traumatic stress disorder (PTSD) [45,46]. PCL-5 has shown good validity and reliability coefficients [43]. The α = 0.96 and the confirmatory factor analysis (CFA) found a good checklist factor structure ($X^{2}$ [161] = 5648.34, *p* < 0.001, an *RMSEA* = 0.077, an *SRMR* = 0.040, a *CFI* = 0.9375, and a *TLI* = 0.924). Blevins et al. [46] reported that the four-factor structure was a model with a good fit ($X^{2}$ [164] = 558.18, *p* < 0.001, a *CFI* = 0.91, a *TLI* = 0.89, an *RMSEA* = 0.07, and an *SRMR* = 0.05; alpha = 0.94), whose optimal score of 31 (out of a total of 80) yielded a sensitivity of 0.77, a specificity of 0.96, an efficiency of 0.93, and a quality of efficiency of 0.73. We used the four-factor structure ([40,46], four scales): re-experiencing, with five items (criterion B, such as repeated, disturbing, and unwanted memories of the stressful experience?); avoidance, with two items (criterion C, such as avoiding memories, thoughts, or feelings related to the stressful experience?); negative alterations in cognition and mood (NACM) with seven items (criterion D, such as Having strong negative beliefs about yourself, other people, or the world (for example, having thoughts such as I am bad, there is something seriously wrong with me, no-one can be trusted, the world is completely dangerous); and hyperarousal with six items (criterion E, such as having difficulty concentrating; see Appendix A). Response options ranged from not at all [0], slightly [1], moderately [2], quite a lot [3], to extremely [4] bothersome symptoms in the past month. The PCL-5 included the less/more-than-a-month-response for how long have the symptoms been bothering you? In addition, the PTSD criterion was considered when a subject selected a two-response option or more for at least one of the B items, one of the C items, two of the D items, and two of the E items, and symptoms had bothered them for over a month.

### 2.3. Procedure

Instructions to participants included the following: The risk of suffering from COVID-19 is an unprecedented social condition that affects us all. The current COVID-19 pandemic is a situation in which we must understand our feelings. We must find out how to deal with them and where to find evidence-based care when required. We therefore invite you to answer the following questionnaire. You will receive feedback on your answers and counseling to help you cope with the emotions, thoughts, and behaviors caused by the current health contingency. Your participation is voluntary, and all the information you provide will be treated confidentially. Your information management will follow Mexican privacy policies for personal data treatment.

### 2.4. Data Analysis

The statistical procedure implicated several steps for the analysis. We examined the dimensionality of the LEC-5, ASSIST, MDE, GA, and PCL-5 scales to provide their construct validity evidence with our sample. We used the CFA from maximum likelihood for continuous variable data and CFA from the diagonally weighted least squares for categorical variables as the estimation methods [32,35]. The overall fit of the models was evaluated using the chi-squared goodness of fit test. Since the chi-squared goodness of fit test is oversensitive to large sample sizes, more emphasis was placed on fit indices, such as the *CFI*, *TLI*, *RMSEA*, and *SRMR.* Specifically, we calculated the Model Optimization Method, the number of free parameters, observations, and missing patterns to validate the models. We used the Model Test User Model with Test statistics, degrees of freedom, *p*-value (chi-squared), and the Model Test Baseline Model. Then, we compared the User Model to the Baseline Model using the CFI and TLI. We calculated the RMSEA with a 90% confidence (interval lower–upper) ≤ 0.05, the SRMR, and their Parameter Estimates with Standard Error and Hessian Observed Information to determine the distance between a hypothesized model and a perfect model. The *SRMR* index was not considered for categorical data as Li [34] recommended. Models with *CFI* and *TLI* values greater than 0.90 and *RMSEA* and *SRMR* values of less than 0.08 and 0.06 were regarded as indicators of good data fit [34,35,36]. We also obtained a Cronbach’s Alpha test for each scale to determine the reliability of the dimensions. Cronbach’s alpha test is based on the item correlation in each scale represented by a number between 0 and 1. A value near to one means that items in the scales are more correlated with each other, evaluating the same latent variable and being stable over time. Cronbach’s alpha makes it possible to determine the degree of independence between the dimensions.

We obtained the scores for each scale and classified subjects who met the violence (LEC-5), AOD (ASSIST), depression (MDE), anxiety (GA), and PTSD (PCL-5) criteria for risk. We calculated groups for polydrug use (more than one harmful AOD use) and comorbidity (more than one group of mental health symptoms). In other words, we obtained the average scores of the scales, and classified participants into At-Risk or Not-at-Risk groups for each dimension. We compared the distribution of the participants by risk level by sex and educational attainment. We therefore performed chi-squared tests, considering *p*-values under 0.05, on participants’ distribution, by groups of risk from violence (victimizing interpersonal and intimate violence, perpetrating interpersonal and intimate violence), harmful AOD use, depression, anxiety, and PTSD symptoms, and by sex or educational attainment of the sample.

We calculated the corresponding relative risks (odds ratios), with their respective 95% confidence intervals, for victimizing interpersonal and intimate violence over harmful AOD use, and harmful AOD use over victimizing and perpetrating interpersonal and intimate violence for the total sample. We also calculated the corresponding relative risks, with the respective 95% confidence intervals, for harmful AOD use and victimizing interpersonal and intimate violence scales over mental health symptoms for the total sample. Odd ratios were calculated through two variables (violence and AODs) with two categories each (at risk and not at risk) and analyzing their contingency table, based on the chi-squared test and *p*-value. Odds ratios have values between 0 and positive infinity. Values around one indicate that one variable poses no risk to the other one. Values over one indicate the likelihood of one variable having risk over the other, while values under one indicate that the variable represents protection against the other variable.

Finally, several tested structural models of the association directionality from victimizing and perpetrating and interpersonal and intimate violence to harmful AOD use and mental health symptoms were run, based on the odds ratio results. We analyzed the predictive models between variables, evaluating the mediating effects of using AOD between violence and mental health symptoms and the moderating effects of the role of sex and educational attainment, with the chi-squared test and their fit indices through the SEM with a mixture of continuous and categorical variables [35]. We represented the final models with good fit indices that proved our hypothesis. All analyses were conducted using Lavaan 0.6–11 in the integrated development environment RSTUDIO^®^ 2022.02.0 from the R Core Team [47] of the Foundation for Statistical Computing. We also used SPSS^®^ 25.0 (IBM Corp., Armonk, NY, USA) [48].

## 3. Results

### 3.1. Confirmatory Factorial Analyses and Cronbach’s Alpha

The results of the factor models of the LEC-5, ASSIST, MDE, GA, and PCL-5 scales are shown in Table 2. Data fitting was adequate, with *CFIs* and *TLIs* > 0.90, *RMSEAs* < 0.08, and *SRMRs* < 0.06. As noted, the categorical CFA indicated a good fit for the four LEC-5 scales: victimizing and perpetrating interpersonal and intimate violence, and the ASSIST-Once in Lifetime AOD Use scale. The CFAs also obtained a good fit for the ASSIST, MDE, GA, and PCL-5 continuum variables, Re-experimentation, Avoidance, NACM, and Hyperactivation. The reliability range of the scales went from 0.60 for the Once in Lifetime Drug Use scale to 0.96 for the Opioid scale from ASSIST. Items within each scale were therefore well-correlated, evaluating latent variables consistently over time. Cronbach’s alpha makes it possible to determine the degree of independence between the dimensions.

### 3.2. Violence, Harmful AOD Use, Depression, Anxiety, and PTSD in the Total Sample and by Sex and Educational Attainment

The distribution of youths at risk for violence, harmful AOD use, depression, generalized anxiety, and PTSD symptom criteria in the total sample and by sex and educational attainment are shown in Table 3. In the overall sample and according to the cutoff score in the corresponding scales, 25.00% of participants were at risk for victimizing interpersonal violence, 25.26% for victimizing intimate violence, 23.48% for perpetrating interpersonal violence, and 15.38% for perpetrating intimate violence. In harmful AOD use, 25.90% of participants were at risk for tobacco use, 20.20% for alcohol use, and 12.50% for cannabis use. Moreover, 18.93% of the total sample were at risk for polydrug use, while 44.46% of participants were at risk for depression, 47.90% for anxiety, and 29.47% for PTSD symptoms. A total of 36.56% of the total sample reported at least two mental health problem (comorbidity).

The percentages of men, women, high school, and university graduates who reported victimizing and perpetrating interpersonal and intimate violence, harmful AOD use, and mental health criteria are also shown in Table 3. Note that the proportions of women at risk for victimizing and perpetrating intimate violence were significantly higher than those of men (*p* < 0.05). The proportion of men at risk for AOD use was significantly higher than that of women (*p* < 0.05), except for sedative use, where women scored higher than men (*p* < 0.05). The proportion of women at risk across mental health conditions was significantly higher than that of men (*p* < 0.05).

There were no significant differences between the proportion of high school participants at risk for any type of violence and those with university degrees (*p* < 0.05). However, for harmful AOD use, participants who had only completed high school were significantly more at risk for tobacco, alcohol, cannabis, cocaine, and stimulant use than those who had completed university (*p* < 0.05). The proportion of high school participants at risk for depression, PTSD, and comorbidity was significantly higher than that of participants with university degrees (*p* < 0.05).

### 3.3. Relative Risks between Violence, Harmful AOD Use, and Mental Health Symptoms

The significant relative risks, with their respective 95% confidence intervals from the odds ratio analysis, are shown in Figure 1 and Figure 2. Remember that values around one indicate that exposure to one variable poses no risk to the other one. Values over one indicate that exposure to one variable affects the other. Victimizing interpersonal and intimate violence predicting harmful AOD use is represented in the upper panel in Figure 1. Participants who had been victims of interpersonal and intimate violence showed increases in harmful use of tobacco, alcohol, cannabis, cocaine, sedatives, hallucinogens, and other drugs (1.353- to 3.153-fold increases). Victimizing intimate violence alone increased stimulant risk by 1.972-fold and polydrug use by 1.417-fold.

Relatively harmful AOD use and victimizing interpersonal and intimate violence risk of perpetrating interpersonal and intimate violence are shown in the lower panel of Figure 1. The risky use of tobacco, alcohol, cannabis, cocaine, stimulants, inhalants, sedatives, hallucinogens, other drugs, and polydrug use increase the perpetration of interpersonal and intimate violence (between 1.456- and 5.233-fold). Victimizing interpersonal and intimate violence resulted in 4.674-fold and 5.539-fold increases in perpetrating interpersonal violence, respectively. Victimizing interpersonal and intimate violence also resulted in 5.512-fold and 6.011-fold increases in perpetrating intimate violence, respectively.

Significant relative risks, with their respective 95% confidence intervals, for harmful AOD use and victimizing interpersonal and intimate violence over mental health symptoms, are shown in Figure 2. The harmful use of tobacco, alcohol, cannabis, sedatives, and polydrug increased depression, anxiety, PTSD, and comorbidity (with 1.448- to 4.436-fold increases). Harmful hallucinogen use predicted depression, anxiety, and comorbidity (with 1.662- to 2.313-fold increases). The harmful use of other drugs predicted depression, anxiety, and PTSD symptoms (with 1.818- to 3.500-fold increases). The harmful use of cocaine and stimulants also predicted depression and anxiety symptoms (with 1.971- to 3.978-fold increases). Victimizing intimate violence predicted depression, anxiety, PTSD symptoms, and comorbidity (with 1.632- to 3.099-fold increases), while victimizing interpersonal violence also predicted depression and anxiety symptoms (with 1.707- to 2.401-fold increases).

### 3.4. Structural Equation Modeling

The best restricted model tested after odds ratios is shown in Figure 3. The final model included paths from victimizing intimate violence to harmful use of tobacco, alcohol, cocaine, sedatives (b_Tob_ = 0.190, b_Alc_ = 0.201, b_Coc_ = 0.204, and b_Sed_ = 0.294, respectively); depression, anxiety, re-experimentation, avoidance (b_MDE_ = 0.233, b_GA_ = 0.200, b_Rex_ = 0.422, and b_Avo_ = 0.140, respectively); and perpetrating interpersonal and intimate violence (b_PIntPV_= 0.204, b_PIntV_ = 0.390, respectively). The model includes a path between harmful sedative use and depression (b_MDE_ = 0.133), and victimizing interpersonal violence and avoidance, NACM, hyperarousal, and perpetrating interpersonal violence (b_Avo_ = 0.242, b_NACM_ = 0.358, b_Hyp_ = 0.319, and b_PIntPV_ = 0.208, respectively). Victimizing intimate violence indirectly affects depression via the risky use of sedatives (combined b_Sed, MDE_ = 0.427). All these path coefficients were significant at *p* < 0.01 or less (e.g., b_Tob_ = 0.190, from victimizing intimate violence to harmful use of tobacco). The model provided a good fit with the data from 204 iterations with 276 parameters (*X*^2^ [2484] = 14,941.17, *p* < 0.001). It resulted in a *CFI* = 0.968, a *TLI* = 0.966, and an *RMSEA* = 0.037 [0.037–0.038]), using a mixture of continuous and categorical observed variables from the total sample. Appendix A shows factor loadings for the observed variables for each scale of the SEM included in Figure 3. In all cases, factor loadings were greater than 0.300.

SEMs by sex and educational attainment samples are shown in Figure 4. Violence scales had to be restricted to obtain models with a good fit. The men’s model considered emotional and sexual abuse items for victimizing intimate violence and physical, emotional, and sexual items for perpetrating interpersonal violence. The women’s model included physical, emotional, and sexual abuse items for victimizing interpersonal violence and physical, emotional, and sexual items for perpetrating interpersonal violence. The high school sample model included physical, emotional, and sexual items for victimizing intimate violence.

The men’s model resulted in a robust predictive path between victimizing intimate violence and the harmful use of tobacco, alcohol, and sedatives, but not cocaine. Men’s SEM showed a strong predictive pattern of victimizing intimate violence for depression, anxiety, re-experimentation, and avoidance symptoms, as well as for perpetrating interpersonal and intimate violence. The men’s model did not confirm victimizing interpersonal violence as a predictor of avoidance, NACM, or hyperarousal or for perpetrating interpersonal violence. Sedative use did not modulate the prediction of violence for depression either. Women’s SEM confirmed the global SEM pattern, except for victimizing interpersonal violence as a predictor of perpetrating interpersonal violence.

Finally, the high school participants’ path model confirmed victimizing intimate violence as a predictor of harmful use of tobacco, alcohol, and cocaine, depression, anxiety, re-experimentation symptoms, and perpetrating intimate violence. However, this path did not predict avoidance, or sedatives, or perpetrating interpersonal violence. Harmful sedative use has separately predicted depression symptoms. High school students’ SEM indicates that victimizing interpersonal violence predicts avoidance, NACM, hyperarousal, and perpetrating interpersonal violence. University students’ SEM replicates all global SEM predictions.

## 4. Discussion

The present study analyzed the relationship between victimizing and perpetrating interpersonal and intimate violence, harmful AODs, and mental health conditions of Mexican youths moderated by sex and educational attainment demographics during the second year of the pandemic. The study validated measurements and models comparing levels of violence, AOD use, depression, anxiety, and PTSD severity during the context of the pandemic. The findings were compared with pre-pandemic and first year of the pandemic prevalence and directionality. Associations between variables were also identified for the entire Mexican youth sample, and by sex and educational attainment.

The findings suggest a good, stable structure of violence, harmful AOD use, depression, anxiety, and PTSD measurements. Latent variables were independent of each other. SEM also proved to be an effective strategy for validating the path between study variables and the odds ratio analysis, the standard type of assessment in these kinds of studies. The valid structure of our variables replicated the conceptualizations of Weathers, Litz et al. [45], Scott-Storey et al. [16], Tiburcio et al. [49], Morales, Robles, Bosch et al. [43], Goldberg et al. [44], and Blevins et al. [46]. The valid structure of the assessment was used as an essential practice as recommended by Elhai and Palmieri [32] and Scott-Storey et al. [16] during emergencies.

A valid structure of victimizing and perpetrating interpersonal and intimate violence in the Mexican youth population comprises behaviors involving physical assault, psychological abuse, sexual assault, and any other unwanted or uncomfortable sexual experiences while victimizing or perpetrating interpersonal or intimate violence, both inside and outside the home. Study participants reported violent behaviors conceptualized by WHO [2], Oram et al. [12], Alexander and Johnson [10], Kourti et al. [11], Scott-Storey et al. [16], and Weathers and Litz et al. [39], providing further information on the second year of the pandemic.

The valid assessment of harmful AOD use has been considered in the WHO definition (2010). Harmful AOD use refers to substances used in the past three months, leading to health, social, legal, and financial problems, failing to conduct what is expected of one and failing to reduce drug use. Friends and relatives also express concern about a person’s use within this conceptualization. Depression, anxiety, and PTSD symptoms were evaluated in accordance with the criteria of the APA [40], Goldberg et al. [44], and Blevins et al. [46].

One in four Mexicans reported interpersonal or intimate violence in the past six months during the second year of the pandemic. These 2021–2022 rates were higher than what White et al. [9] reported in 2012–2020, below what Glowacz et al. [17] reported in 2020, and supported Kourti et al.’s [11] proposal that intimate violence increased in the first year of the pandemic. Our findings also identified the perpetration of interpersonal and intimate violence during the second year of the pandemic: 23.48% of Mexican youth reported perpetrating interpersonal violence, while 15.38% reported perpetrating intimate abuse. The Mexican community has therefore reported high levels of violence as White et al. [9] detailed when comparing their results with clinical groups. Glowacz et al. [17] also proposed that younger participants involved in a relationship, as in the case of the Mexican youth in the study, were more likely to experience and perpetrate physical and psychological violence during lockdown.

Young Mexican men and women also reported victimizing and perpetrating interpersonal violence. However, more women than men reported suffering and perpetrating intimate violence, partially contrasting with Glowacz et al.’s [17] report that more men than women suffer from physical intimate violence but supporting Glowacz et al.’s [17] conclusion that more women suffer from psychological intimate violence. The fact that similar proportions of men and women suffered and perpetrated interpersonal violence and that different proportions by gender suffered and perpetrated intimate violence supports the pre-pandemic findings of Scott-Storey et al. [16]. They reported that it is possible to observe symmetric violence between the sexes, together with asymmetries related to the forms of intimate violence. Scott-Storey et al. [16] stated that, although fewer men may asymmetrically report victimizing intimate violence, they are used to perceiving it in a context where emotional and sexual abuse happens inside families. Our study suggests that men’s victimizing intimate violence contains emotional and sexual forms of violence, asymmetrically by gender. Violence was asymmetric between the sexes for victimizing and perpetrating intimate violence, but symmetric between the sexes for victimizing and perpetrating interpersonal violence. Moreover, victimizing intimate violence was asymmetric between the sexes since it is exclusively referred to as emotional and sexual violence for men. The findings show violence between and within the sexes for our sample during the second year of the pandemic. Forms of violence were symmetric by educational attainment in the study in 2021–2022.

Mexican youths reporting violence also mentioned harmful AOD use, depression, anxiety, and PTSD symptomatology in 2021 and 2022. A total of 25.90% reported harmful tobacco use, 20.20% reported harmful alcohol use, and 18.93% reported using more than two psychoactive substances. This coincides with Craig et al.’s [19] pre-pandemic findings that one in five adolescents engaged in regular drug use. Our findings of harmful AOD use appeared to be below global prevalence based on the only epidemiologic study in Mexico by NCA-MHSAMOS [8] of a 2020 sample. Note, however, that the Mexican youths in our study reported harmful use of AODs rather than prevalence. Harmful use means that people require long-term treatment due to their use of AODs. In this context, 12.50% of young people were harmfully using cannabis, 1.80% cocaine, and 4.80% sedatives. The need for long-term treatment was evident in 2021 and 2022. The study also found more men abusing tobacco, alcohol, cannabis, cocaine, and polydrugs than women (29.50%, 24.20%, 15.80%, 2.70%, and 22.69% versus 24.20%, 18.60%, 11.00%, 1.50%, and 17.23%, respectively). More women, however, were abusing sedatives than men (5.40% versus 3.50%, respectively). The findings also indicate that those who had only completed high school abused tobacco, alcohol, cannabis, and cocaine more than participants with a university degree (29.50%, 22.00%, 14.20%, and 2.80% versus 24.80%, 19.60%, 12.00%, and 1.60%, respectively). The findings suggest the symmetrical abuse of sedatives and polydrug use by educational attainment in 2021–2022.

The rates of mental health problems among Mexican youth were also high during the second year of the pandemic: depression (44.46%), generalized anxiety (47.90%), and PTSD (29.47%). Our proportions were slightly below those that Craig and collaborators [19] found with adolescents in 2020, but above what Bourmistrova et al. [6] suggested as consequences: mental health symptoms after recovery from illness. Our study also indicated that 36.56% suffered from comorbid mental health symptoms in 2021 and 2022 and observed asymmetries between sex and educational attainment. More women reported depression, anxiety, PTSD, and comorbid symptoms than men (47.61%, 50.40%, 31.77%, and 39.27% versus 34.51%, 42.30%, 24.42%, and 30.60% respectively). More Mexican youths who had only completed high school also reported depression, PTSD, and comorbid symptoms than participants with a university degree (47.01%, 33.51%, and 40.56% versus 43.71%, 28.28%, and 35.39%, respectively).

The proportion of young people suffering or perpetrating violence, using AODs, and having mental health problems can be explained by conditions during the pandemic. Glowacz et al. [17] and Kourti et al. [11] have suggested that lockdown or losses during the pandemic could explain these circumstances. Future research, however, could explore how sociodemographic settings related to the pandemic, such as social distance, loss of loved ones, and losing jobs, were related to violence, drug use, and mental health illness in the second year of the pandemic. Meanwhile, validating the structure of the variables paved the way for path analysis and proposals for future research. Describing the directionality of links between the variables could contribute to future research and prevent certain conditions in future pandemics.

The hypothesis of the study focused on the relationship between violence, harmful AOD use, and mental health illness. The odds ratios suggested how these conditions were related in 2021 and 2022. Victimizing interpersonal and intimate violence increased harmful use of tobacco, alcohol, and sedative use (1.353- to 3.153-fold), above the pre-pandemic amount proposed by Brabete et al. [20] and Machisa and Shamu [21]. Victimizing interpersonal violence led to a 1.707- to 2.401-fold increase in depression and anxiety symptoms in Mexican youths—as in the odds ratios reported by Craig et al. [19] in 2020 and by White et al. [9] pre-pandemic. Victimizing intimate violence also predicted depression, anxiety, PTSD, and comorbid symptomatology (causing 1.632- to 3.099-fold increases) in 2021 and 2022.

The odds ratios also suggested that harmful AOD use predicted perpetrating intimate violence together with mental health conditions. All types of harmful use of AODs increased the perpetration of both interpersonal and intimate violence 1.456- to 5.233-fold as Glowacz et al. [17], Brabete et al. [20], and Caldentey et al. [23] suggested before and in the first year of the pandemic. The risk of perpetrating violence seemed to vary according to the drug used, as Zhong et al. [27] found in the pre-pandemic era. Odds ratios also suggest that using drugs and victimizing intimate violence are associated with poor mental health, as Bosch et al. [28] and Brabete et al. [20] suggested before the pandemic. The harmful use of tobacco, alcohol, cannabis, sedatives, and polydrugs increased depression, anxiety, PTSD, and comorbidity 1.448 to 4.436-fold in the second year of the pandemic. Harmful cocaine and stimulant use increased depression and anxiety 1.971- to 3.978-fold. The harmful use of hallucinogens predicted depression, anxiety, and comorbidity (with 1.662- to 2.313-fold increases). The harmful use of other drugs predicted depression, anxiety, and PTSD symptoms (with 1.818- to 3.500-fold increases). Finally, the odds ratio also suggests that victimizing interpersonal and intimate violence resulted in a 4.674- to 6.011-fold increase in perpetrating interpersonal and intimate violence.

The odds ratio pointed to a clear association between violence, harmful AOD use, and mental health conditions. The global predictive model was therefore based on the resulting odds ratio. The SEM analysis suggested that victimizing intimate violence exclusively predicted the harmful use of tobacco–alcohol–cocaine–sedatives (Ha1), depression (Ha2), generalized anxiety (Ha3), and re-experimentation and avoidance symptomatology from the Post-Traumatic Stress Disorder (PTSD; Ha4) with our Mexican youth sample in 2021 and 2022. Victimizing intimate violence related to drug use and mental health conditions supports this association, as described by Brabete et al. [20], Machisa and Shamu [21], Craig et al. [19], Glowacz et al. [17], and White et al. [9] both pre-pandemic and in 2020.

Victimizing interpersonal violence did not predict mental health symptoms in our youth sample—as Craig et al. [19] and White et al. [9] suggested pre-pandemic and in 2020. Harmful AOD use did not predict perpetrating interpersonal or intimate violence (Ho5), anxiety (Ho7), or PTSD symptoms (Ho8), as suggested by the odds ratio, and Glowacz et al. [17], Brabete et al. [20], and Caldentey et al. [23] reported pre-pandemic and in the first year of the emergency. The model did, however, suggest a predictive path between harmful use of sedatives and depression (Ha6). The use of sedatives seemed to mediate victimizing intimate violence and depression in young Mexicans in 2021 and 2022, supporting the findings of Bosch et al. [28] and Brabete et al. [20] before and in 2020.

The global SEM also notes that victimizing interpersonal and intimate violence predicted perpetrating interpersonal violence, and that victimizing intimate violence predicted perpetrating intimate abuse by Mexican youth during the second year of the pandemic. Both victimizing and perpetrating interpersonal and intimate violence have been reported, not just victimizing intimate abuse as White et al. [9] noted pre-pandemic.

The study suggested asymmetric path models associated with sex and education. Victimizing intimate violence was identified as the main predictor of harmful use of AODs, depression, anxiety, re-experimentation, avoidance, and perpetrating violence in the women’s model, as Biswas [29], Hernandez [30] and Gubi et al. [31] suggested pre-pandemic. Victimizing interpersonal violence only predicted more severe PTSD symptomatology, such as negative alterations in cognition and mood, and hyperarousal plus avoidance. Victimizing both interpersonal and intimate violence were independent paths for the young women’s sample, representing normal and complex PTSD symptomatology related to each form of violence [50].

One hypothesis about the complexity of PTSD is related to forms of violence. Keely et al. [50] have suggested that complex PTSD can be described as pervasive problems with affect regulation (NACM), persistent beliefs about oneself as diminished (succumbing to adverse circumstances), persistent difficulties in sustaining relationships (feeling close to others), and disturbances causing significantly impaired functioning. These severe symptoms were reported by young Mexican women when they experienced interpersonal violence rather than intimate abuse. Victimizing intimate violence was related to normal PTSD symptoms in addition to depression, anxiety, and drug use, as a possible means of coping. Future longitudinal research could analyze complex PTSD related to forms of violence. It is essential, however, to study this relationship in a context where coping skills could halt the progression of acute stress symptoms to complex PTSD.

The study indicated a dense men’s path solely based on victimizing intimate violence. Victimizing intimate violence strongly predicted harmful tobacco–alcohol–sedative use, depression, anxiety, re-experimentation, and avoidance—normal PTSD symptoms [50]. Victimizing intimate violence also predicted perpetrating intimate violence by young men. Additional research should confirm whether a violence-escalating mechanism can occur once young people have been interpersonally or intimately victimized [16]. Glowacz et al. [17] and Scott-Storey et al. [16] have both suggested asymmetries in violence models based on a person’s sex. Our study suggests a particular role of victimizing intimate violence in men’s model relationships.

Scott-Storey and collaborators [16] have already proposed that asymmetries in victimizing intimate violence by sex may result from differences in the perception of violence by sex in a context of social inequities, and normalized violation of human rights as occurs in Mexico [51]. Both men and women suffer intimate violence. However, forms of violence and their consequences may differ by sex due to the patriarchal culture and the role of power and control in societies. Although men disclose the forms of violence suffered, they seem to view emotional and sexual abuse as more dangerous than physical violence. Women can endure several forms of violence for long periods of time, suffering greater consequences [51]. Men and women, however, seem to cope with victimizing intimate violence through AOD use [52]. Future longitudinal research should therefore address forms of violence, gender interaction, and the consequences of perceived abuse by sex and culture in several low-income countries where human rights are routinely violated.

The high-school predictive model is split into three paths. Victimizing intimate violence predicted harmful use of tobacco–alcohol–cocaine, depression, anxiety, re-experimentation, and perpetrating intimate violence. Victimizing interpersonal violence predicted avoidance, negative alterations in cognition–mood, hyperarousal, complex PTSD, and perpetrating interpersonal violence. The last pattern indicates that the harmful use of sedatives predicts depression symptoms in a high school sample. Hernandez [30], Gubi et al. [31], and Dos-Santos et al. [25] suggested that educational attainment predicted violence based on pre- and first year of pandemic findings, but our study presented single paths associated with forms of violence and sedative use. The findings constitute a baseline to explore the hypothesis of the mechanisms behind these paths. Hernandez [30], Gubi et al. [31], and Dos-Santos et al. [25] suggested that lower educational attainment predicts violence, while Craig et al. [19] reported that the age of onset of drug use is lower in adolescents with a lower educational attainment. However, the mechanisms in the paths of the model for participants with lower educational attainment could be addressed in future longitudinal studies. Another hypothesis concerns the role of drug use as a self-medicating mechanism. Future longitudinal research could determine whether the use of tobacco, alcohol, cocaine, and sedatives is a coping mechanism to numb feelings related to violence or to avoid thinking about experiencing violence.

Mexican youths with university degrees were extensively represented in the global violence–AODs–mental-health predictive model. In general, participants showed no relationship between AOD use and perpetrating intimate or interpersonal violence. Our Mexican sample may be too young to show that drug use predicts the perpetration of violence as Ismayilova [53] predicted with a sample of older participants. The association between drug use and perpetrating interpersonal and intimate violence may be linked to being older, several life conditions, being a caregiver, having a lower educational attainment, or experiencing certain socioeconomic conditions. Future research could therefore explore these conditions that could explain the associations between AOD use and violence perpetrated as Islam et al. [54] proposed.

The present study examined the association between victimizing and perpetrating interpersonal and intimate violence, harmful AOD use, and the mental health conditions of Mexican youths, moderated by sex and education demographics, in 2021 and 2022. A valid measurement of variables suggested that Mexican youths reported higher levels of violence than those reported before the pandemic, but in a culturally different sample from a low-middle-income country (LMIC). Harmful AOD use rates were similar to pre-pandemic levels, whereas mental health symptomatology was lower than that reported in 2020 research. The path model with a good fit has also suggested that victimizing intimate violence predicted harmful drug use and perpetrating intimate violence. Both victimizing both intimate and interpersonal violence has predicted mental health symptomatology and perpetrating interpersonal abuse. Mexican young people were asymmetrically distributed by gender for victimizing and perpetrating intimate violence, but symmetrically distributed for victimizing and perpetrating interpersonal abuse. Forms of victimizing intimate violence by sex were asymmetrically observed by sex due to men’s reports of emotional and sexual abuse. A strong path of victimizing intimate violence following drug use, mental health symptomatology, and perpetrating violence was observed for men. There were two patterns of violence for women: one linked to victimizing intimate violence, predicting drug use, mental health symptoms, and perpetrating violence, and another to victimizing interpersonal violence predicting severe PTSD symptomatology. Young people who had only completed high school showed three predictive patterns: one for victimizing intimate violence, another for victimizing interpersonal violence, and yet another for harmful sedative use. Young people with a university degree yielded a broad model with all the patterns interacting as in the global predictive model.

The global comprehensive and associative models of the study helped describe violence, drug use, and mental health relationships, laying the groundwork for future research on the mechanisms underlying predictive patterns in LMIC. Explaining these mechanisms could help with the design of more cost-effective preventive programs and public policies and suggest how to cope with mental health conditions during emergencies in the community context. The Mexican government could design strategies to prevent young people from experiencing interpersonal and intimate violence, preventing harmful AOD use, and mental health issues.

## 5. Conclusions

The present study analyzed the relationships between violence, harmful drug use, and mental health conditions in Mexican youths, including social determinants, such as sex and academic achievement during the second year of the pandemic (2021–2022). Young Mexicans suffered from intimate violence, perpetrated it, harmfully used AODs, and presented mental health symptomatology. The levels of interpersonal and intimate violence were above those reported in other studies before and during the first year of the pandemic. The study widely described victimizing and perpetrating interpersonal and intimate violence. The findings suggest asymmetric victimizing and perpetrating intimate violence, and symmetric victimizing and perpetrating interpersonal abuse between the sexes. Symmetries were observed in all forms of violence between young people by educational attainment.

AOD use was reported in the pre-pandemic period, but this study found high proportions of multiple use of harmful drugs in 2021 and 2022. More young men were harmfully using drugs, except for sedatives, for which women were at a higher risk. More young people who had completed high school harmfully use tobacco, alcohol, cannabis, cocaine, and stimulants than those with university degrees. Mental health symptoms were below those reported during the first pandemic wave in 2020, but above those cited as sequels of COVID-19. There were asymmetries in depression, anxiety, and PTSD between gender, which affected women more than men, and in depression, anxiety, and comorbidity between educational attainment levels, which were more prevalent in those who had only completed high school than in those who had completed university degrees.

Predictive models indicated that being a victim of intimate violence predicts harmful use of tobacco, alcohol, cocaine, and sedatives, depression, generalized anxiety, re-experimentation, avoidance, and perpetrating interpersonal and intimate violence. Being a victim of interpersonal violence also results in severe PTSD symptoms, such as avoidance, negative alterations in cognition and mood, and hyperarousal signs. Harmful sedative use also predicts depression. Harmful drug use, however, did not predict perpetrating interpersonal, intimate violence, anxiety, or PTSD symptoms in Mexican youths during the second year of the pandemic.

The findings suggest that being a victim of interpersonal and intimate violence resulted in perpetrating interpersonal and intimate abuse. The findings also indicated asymmetric predictive models between sex and educational attainment levels. Men reported intimate violence, both emotional and sexual, closely linked to harmful use of tobacco, alcohol, sedatives, depression, anxiety, normal PTSD symptoms, re-experimentation and avoidance, and perpetrating interpersonal and intimate violence. Women reported the same experience of intimate violence path, which also includes physical abuse, harmful cocaine use, and the mediating role of sedatives with depression. The women’s model also included the victimizing interpersonal violence path associated with the symptomatology linked to complex PTSD reported by Keeley et al. [50]. Three separate paths characterized the high school youth model. One involves victimizing intimate violence, another involves victimizing interpersonal violence, and yet another involves sedative use predicting depression. Young Mexicans with a university degree yielded a broad model with all the patterns interacting as in the global predictive model.

The findings of this study underscore the importance of preventing intimate violence as part of public health approaches to prevent harmful drug use and mental health conditions. Early interventions can provide care for both victims and perpetrators and halt the escalation and inflicting of more severe interpersonal and intimate violence. The conclusions also point to the importance of examining the role of unequal gender roles, social determinants, and drug use as a mechanism for coping with violence and mental health symptoms.

It is essential for first responders and healthcare providers to provide knowledge on mechanisms to reduce violence, drug use, and mental health problems. Interpersonal and intimate violence will continue unless information on the importance of changing gender norms, roles, and attitudes that perpetuate abuse is provided. Health care providers should offer timely, coordinated services to tackle violence, substance use, and mental illness. Public policies should provide programs to reduce gender inequities, increase community empowerment, and promote justice and human rights.

## 6. Limitations

This is a cross-sectional study. Despite the use of advanced statistical analysis, such as the SEM, which is extremely reliable, the association between violence, drug use, and mental health conditions should be explored through longitudinal studies. The latter contribute to confirming and understanding the mechanisms underlying the effects of intimate violence, substance dependence, depression, anxiety, and PTSD symptom development.

The findings should also be considered in the context of screening. This means that, although measures of variables were validated, our strategy does not constitute a diagnosis of mental health or substance use disorders. Screening at the community level over-estimates symptoms and reports [44]. Future studies should evaluate the consistency between screening and diagnosis, including the sensitivity and specificity of psychometric tools to empirically confirm the proposed model.

Moreover, subsequent research should consider verifying the processes that explain how social determinants are related to our model, such as family size, family interaction, lockdown, and the physical illness of caregivers during a health emergency. Considering these factors and their mediating or moderating effect between violence, AOD use, and mental health conditions could contribute to a better understanding of these socio-demographic and socially relevant worldwide conditions. Moreover, future studies should identify biased sources in the items as Morales, Robles, Bosch et al. [43] have already reported between sex and educational attainment to increase the accuracy of comparisons.

Finally, future studies should consider improving the representativeness of the Mexican youth sample because participants in the current study voluntarily chose to participate. This would contribute to improving the design of practical, effective preventive measures and interventions in Mexico.

## Figures and Tables

**Figure 1 ijerph-20-06484-f001:**
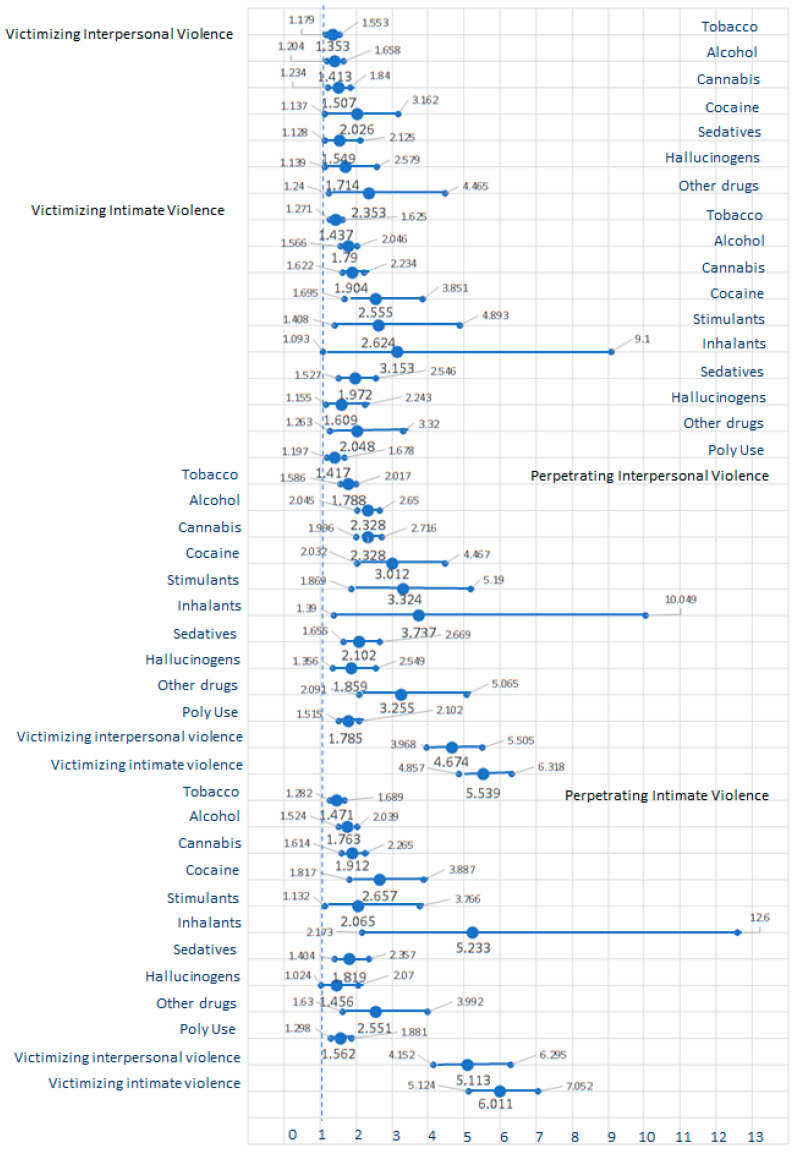
Relative risks, with their respective 95% confidence intervals. Variables on the left column predict the variables listed on the right side of the graph (dependent variables). The upper half of the graph shows the effect of victimizing interpersonal and intimate violence on harmful AOD use. The bottom half of the graph shows the effect of AOD and victimizing interpersonal and intimate violence over perpetrating interpersonal and intimate violence.

**Figure 2 ijerph-20-06484-f002:**
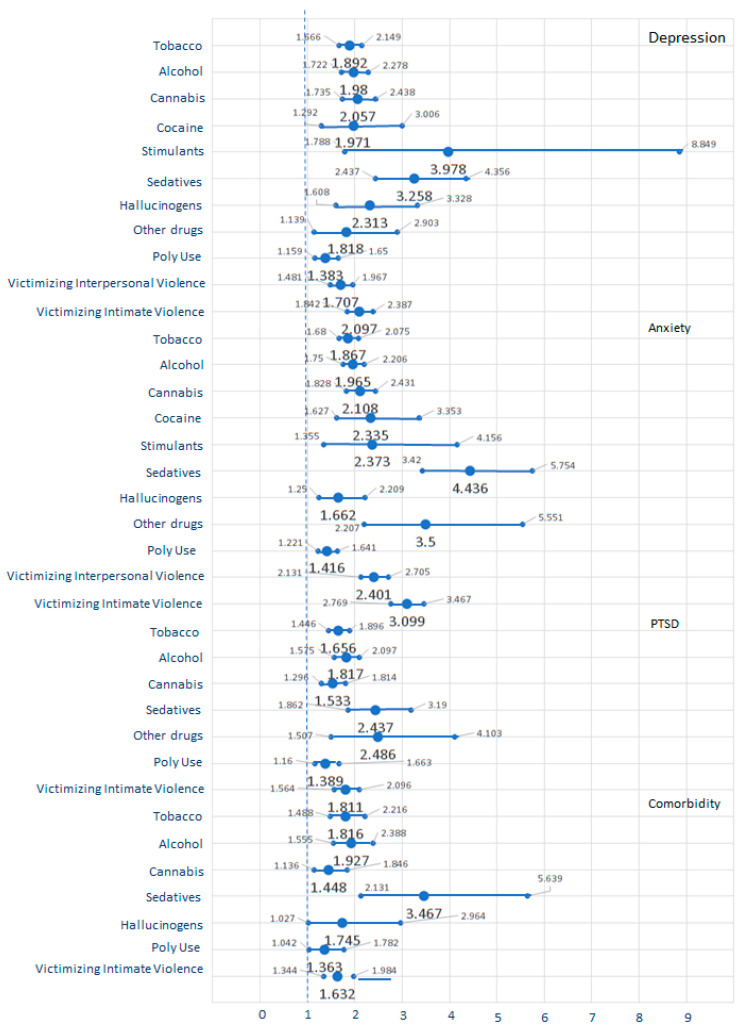
Relative risks, with their respective 95% confidence intervals. Variables on the left column predict variables named on the right side of the graph (dependent variables). Graph shows harmful AOD use and victimizing interpersonal and intimate violence affecting mental health symptoms for the total sample.

**Figure 3 ijerph-20-06484-f003:**
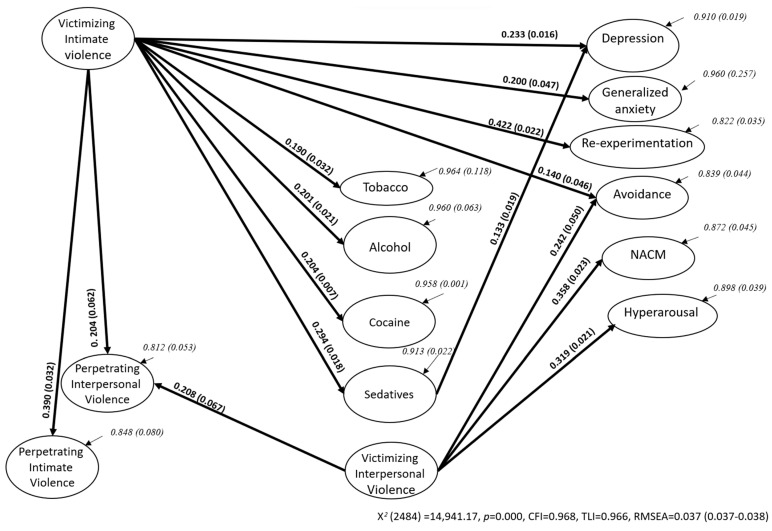
Variables from SEM, path coefficients, and residual variances for the whole sample.

**Figure 4 ijerph-20-06484-f004:**
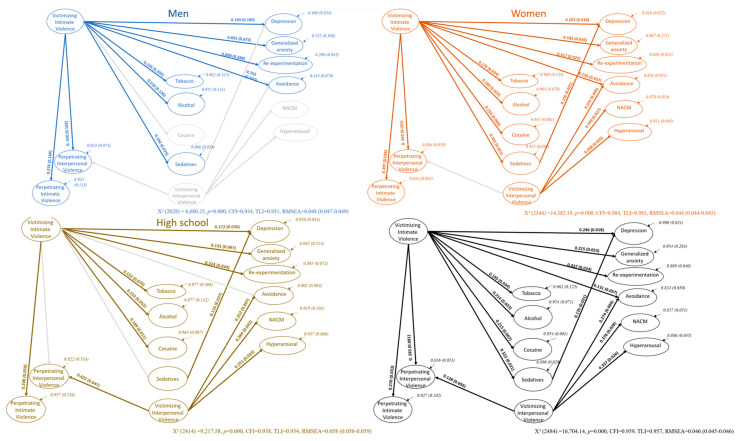
Sex and educational attainment SEMs with path coefficients and residual variances.

**Table 1 ijerph-20-06484-t001:** Distribution of the participants by sex and educational attainment.

**TOTAL**							
*n*				%			
7420				100			
**Men**		**Women**		**High school**		**University**	
*n*	*%*	*n*	*%*	*n*	*%*	*n*	%
2314	31.20	5106	68.8	1689	22.80	5731	77.20

**Table 2 ijerph-20-06484-t002:** Chi-squared analysis, degrees of freedom, *p*-values, fit indices, and Cronbach’s alpha by scales for the total sample.

Scales-Factors	*X* ^2^	*df*	*p* ≤	*RMSEA*	*Confidence Interval*	*SRMR*	*CFI*	*TLI*	*Cronbach’s Alpha*
**LEC-5**									
Victimizing Interpersonal Violence	343.566	13	0.001	0.059	0.053–0.064		0.949	0.917	0.76
Victimizing Intimate Violence	6.087	2	0.048	0.017	0.001–0.032		0.999	0.997	0.76
Perpetrating Interpersonal Violence	94.634	2	0.001	0.079	0.066–0.093		0.952	0.855	0.68
Perpetrating Intimate Violence	0.000	0	0.000	0.000	0.000–0.000		1.000	1.000	0.68
**ASSIST**									
Once in Lifetime	272.565	35	0.001	0.030	0.027–0.034		0.981	0.975	0.60
Tobacco	86.199	4	0.001	0.053	0.043–0.063	0.013	0.994	0.986	0.83
Alcohol	163.646	8	0.001	0.051	0.045–0.058	0.018	0.986	0.973	0.77
Cannabis	155.431	7	0.001	0.053	0.046–0.061	0.013	0.992	0.982	0.85
Cocaine	190.925	6	0.001	0.064	0.057–0.072	0.014	0.992	0.981	0.87
Stimulants	161.657	6	0.001	0.059	0.051–0.067	0.008	0.995	0.988	0.92
Inhalants	65.051	3	0.001	0.053	0.042–0.064	0.010	0.997	0.985	0.85
Sedatives	94.670	7	0.001	0.041	0.034–0.049	0.010	0.995	0.990	0.86
Hallucinogens	323.535	8	0.001	0.073	0.066–0.080	0.026	0.960	0.925	0.71
Opioids	158.546	3	0.001	0.084	0.073–0.095	0.009	0.998	0.990	0.96
Other	220.741	7	0.001	0.064	0.057–0.072	0.017	0.989	0.976	0.86
**MDE**									
Depression	1210.246	42	0.001	0.072	0.069–0.076	0.034	0.954	0.939	0.89
**GA**									
Anxiety	74.940	5	0.001	0.043	0.035–0.052	0.005	0.998	0.995	0.93
**PCL-5**									
Re-experimentation	15.073	4	0.005	0.024	0.012–0.038	0.008	0.999	0.997	0.86
NACM	229.317	10	0.001	0.068	0.061–0.076	0.020	0.983	0.964	0.86
Hyperarousal	200.198	8	0.001	0.071	0.063–0.080	0.030	0.972	0.947	0.78
PTSD	4535.593	162	0.001	0.076	0.074–0.078	0.044	0.916	0.901	0.93

Note. Test: LEC-5 = Life Events Checklist; ASSIST = Alcohol, Smoking, and Substance Involvement Screening Test; MDE = Major Depressive Episode; GA = Generalized Anxiety; PCL-5 = Post-traumatic Checklist. Scales: NACM = Negative Alterations in Cognition and Mood; PTSD = Post-traumatic Stress Disorder (including the two avoidance items). Categorical Variables do not show SRMR as Li (35) recommended.

**Table 3 ijerph-20-06484-t003:** Percentage of youths according to violence, harmful AOD use, depression, and anxiety, and PTSD criteria for the total sample and by sex and educational attainment.

**TOTAL**															
*n*								%							
7420								100							
**Victimizing Interpersonal Violence**								**Victimizing Intimate Violence**							
*n*				%				*n*				%			
1853				25.00				1874				25.26			
**Men**		**Women**		**High school**		**University**		**Men**		**Women ***		**High school**		**University**	
*n*	%	*n*	%	*n*	%	*n*	%	*n*	%	*n*	%	*n*	%	*n*	%
579	25.02	1274	24.95	407	24.10	1446	25.23	515	22.26	1359	26.62	427	25.28	1447	25.25
**Perpetrating Interpersonal Violence**								**Perpetrating Intimate Violence**							
*n*				%				*n*				%			
1742				23.48				1141				15.38			
**Men**		**Women**		**High school**		**University**		**Men**		**Women ***		**High school**		**University**	
*n*	%	*n*	%	*n*	%	*n*	%	*n*	%	*n*	%	*n*	%	*n*	%
532	22.99	1210	23.70	384	22.73	1358	23.70	315	13.61	826	16.18	253	14.98	888	15.49
**Harmful tobacco use**								**Harmful alcohol use**							
*n*				%				*n*				%			
1919				25.90				1497				20.20			
**Men ***		**Women**		**High school ***		**University**		**Men ***		**Women**		**High school ***		**University**	
*n*	%	*n*	%	*n*	%	*n*	%	*n*	%	*n*	%	*n*	%	*n*	%
683	29.50	1236	24.20	498	29.50	1421	24.80	547	23.60	950	18.60	371	22.00	1126	19.60
**Harmful cannabis use**								**Harmful cocaine use**							
*n*				%				*n*				%			
927				12.50				137				1.80			
**Men ***		**Women**		**High school ***		**University**		**Men ***		**Women**		**High school ***		**University**	
*n*	%	*n*	%	*n*	%	*n*	%	*n*	%	*n*	%	*n*	%	*n*	%
365	15.80	562	11.00	239	14.20	688	12.00	62	2.70	75	1.50	48	2.80	89	1.60
**Harmful stimulant use**								**Harmful inhalant use**							
*n*				**Total** %				*n*				%			
57				0.80				22				0.30			
**Men ***		**Women**		**High school ***		**University**		**Men**		**Women**		**High school**		**University**	
*n*	%	*n*	%	*n*	%	*n*	%	*n*	%	*n*	%	*n*	%	*n*	%
26	1.10	31	0.60	22	1.30	35	0.60	10	0.40	12	0.20	8	0.50	14	0.20
**Harmful sedative use**								**Harmful hallucinogen use**							
*n*				%				*n*				%			
357				4.80				203				2.70			
**Men**		**Women ***		**High school**		**University**		**Men ***		**Women**		**High school**		**University**	
*n*	%	*n*	%	*n*	%	*n*	%	*n*	%	*n*	%	*n*	%	*n*	%
81	3.50	276	5.40	84	5.00	273	4.80	85	3.70	118	2.30	56	3.30	147	2.60
**Harmful opioid use**								**Harmful use of other drugs**							
*n*				%				*n*				%			
8				0.10				100				1.30			
**Men**		**Women**		**High school**		**University**		**Men**		**Women**		**High school**		**University**	
*n*	%	*n*	%	*n*	%	*n*	%	*n*	%	*n*	%	*n*	%	*n*	%
3	0.10	5	0.10	2	0.10	6	0.10	32	1.40	68	1.30	25	1.50	75	1.30
**Polydrug Use**															
*n*				%											
1405				18.93											
**Men ***		**Women**		**High school**		**University**									
*n*	%	*n*	%	*n*	%	*n*	%								
525	22.69	880	17.23	352	20.84	1053	18.37								
**Depression**								**Generalized Anxiety**							
*n*				%				*n*				%			
3299				44.46				3553				47.90			
**Men**		**Women ***		**High school ***		**University**		**Men**		**Women ***		**High school**		**University**	
*n*	%	*n*	%	*n*	%	*n*	%	*n*	%	*n*	%	*n*	%	*n*	%
868	37.51	2431	47.61	794	47.01	2505	43.71	979	42.30	2574	50.40	834	49.40	2719	47.40
**PTSD symptoms**								**Comorbidity**							
*n*				%				*n*				%			
2187				29.47				2713				36.56			
**Men**		**Women ***		**High school ***		**University**		**Men**		**Women**		**High school ***		**University**	
*n*	%	*n*	%	*n*	%	*n*	%	*n*	%	*n*	%	*n*	%	*n*	%
565	24.42	1622	31.77	566	33.51	1621	28.28	708	30.60	2005	39.27	685	40.56	2028	35.39

Note. * Significant differences between groups < 0.05 according to the chi-squared analysis. At-risk groups were included in the instruments section. This means: (1) a score over one for victimizing perpetrating, interpersonal, or intimate violence; (2) a score over four for all drugs except alcohol, for which the score has to be over 11 to meet the criteria; (3) more than one harmful AOD use for polydrug use; (4) meeting criteria A and B for depression; (5) an average of over 60% for anxiety; (6) a two -response option or more for at least one of the B items, one of the C items, two of the D items, two of the E items, and bothersome symptoms for over a month for PTSD; and (7) more than one group of mental health symptoms for comorbidity.

## Data Availability

The original contributions presented in the study are included in the article; further inquiries should be sent to the corresponding author.

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
