# Peer review of "Interpersonal and Intimate Violence in Mexican Youth: Drug Use, Depression, Anxiety, and Stress during the COVID-19 Pandemic"

_ijerph, 2023, doi:10.3390/ijerph20156484_

Round 1
Reviewer 1 Report
The introduction is well-organized, and the flow of information is logical. However, it could benefit from clearer section headings or subheadings to improve readability and facilitate navigation through the text.
The introduction effectively justifies the need for the study by highlighting the gaps in the existing literature, such as the unclear directionality of the relationships between interpersonal violence, AOD use, and mental health conditions during the pandemic. Additionally, the inclusion of specific data from Mexico further strengthens the rationale for conducting the study in this context.
The inclusion of studies conducted during the COVID-19 pandemic, as well as pre-pandemic research, helps establish a comprehensive understanding of the topic. However, it would be beneficial to briefly mention any limitations or gaps in the existing studies to further emphasize the need for the current research.
The overall language and clarity of the introduction are good. However, there are a few sentences that could be rephrased for better readability and understanding. Additionally, proofreading for minor grammatical errors and inconsistencies would further improve the quality of the manuscript.
Methods
The methods section provides detailed information about the study design, participants, instruments, and procedures. Overall, the section is well-structured and provides sufficient information to understand the study methodology. However, there are a few areas where the description could be further improved for clarity and completeness.
The section starts with a clear description of the number of participants surveyed and their demographic characteristics. However, it would be helpful to include additional information about the recruitment process and any inclusion/exclusion criteria, if applicable. This would provide a better understanding of how the sample was obtained and ensure transparency in participant selection.
The section briefly mentions the process of obtaining informed consent, but it would be beneficial to provide more details. Specifically, it would be helpful to explain how participants were informed about the purpose of the study, the voluntary nature of participation, and their rights regarding data confidentiality and withdrawal. Additionally, it would be useful to mention whether any ethical approval was obtained for the study and provide the name of the approving institution.
The section describes the instruments used in the study, namely the Life Events Checklist 5th Edition (LEC-5), WHO Alcohol, Smoking, and Substance Involvement Screening Test (ASSIST), Major-Depressive-Episode (MDE) checklist, Generalized Anxiety (GA) scale, and Posttraumatic Checklist (PCL-5). While the instruments are briefly explained, it would be beneficial to provide more information about the psychometric properties of these instruments, such as their reliability and validity coefficients. Additionally, it would be useful to mention the specific scales or subscales used within each instrument and provide a brief rationale for their selection.
The section describes the procedure followed in the study, including the invitation of participants and data collection period. However, it would be helpful to provide more information about how participants accessed the web-based application and whether any reminders were sent to increase response rates. Additionally, it would be beneficial to explain how the feedback and counseling were provided to participants and what resources were made available to them.
To enhance the study further, it would be beneficial to consider the potential moderating or mediating effects of other factors
Results:
The authors present the results in a clear and organized manner, making it easy to understand the key findings of the study. However, there are a few suggestions and points that need to be addressed to improve the clarity and presentation of the results.
The authors mention that the data fitting was adequate, with CFIs and TLIs > 0.90, RMSEAs < 0.08, and SRMRs < 0.06. It would be helpful if they provide a brief explanation of these fit indices for readers who may not be familiar with them.
In Table 2, where the results from the factor models are presented, it would be beneficial to provide more information about the specific factors or latent variables included in each scale. This would help readers understand the underlying constructs being measured by the different scales.
The reliability range of the scales is mentioned, but it would be valuable to include more details about the specific reliability measures used (e.g., Cronbach's alpha) and their interpretation in the context of the study. Additionally, it would be helpful to provide a brief discussion of the implications of the reliability estimates for the interpretation of the scale scores.
Table 3 provides information about the distribution of participants at risk for various outcomes, such as violence, harmful AOD use, depression, anxiety, and PTSD symptoms. While the table is informative, it would be beneficial to include additional information on the sample size and the specific criteria used to define "at-risk" status for each outcome. This would help readers understand the magnitude and clinical significance of the reported percentages.
The authors discuss significant differences between groups (e.g., men vs. women, high school vs. university graduates) in terms of the proportions at risk for different outcomes. However, it is not clear how these differences were assessed statistically. Providing information about the statistical tests used and the significance levels would enhance the rigor and transparency of the analysis.
Lastly, the authors mention that all path coefficients in the models were significant at p < 0.01 or less. While this information is important, it would be helpful to include the actual values of the path coefficients to provide a quantitative understanding of the relationships between variables.
Some minor changes
Author Response
We really appreciate the valuable comments. Here we write the corrections made. Thank you very much.
Corrections cover letter
Reviewer 1
- The introduction is well-organized, and the flow of information is logical. However, it could benefit from clearer section headings or subheadings to improve readability and facilitate navigation through the text.
Thanks for this observation! We include subheadings to improve readability and facilitate the reading..
“1.1. Violence, AOD use, and mental health conditions before and in the first year of the pandemic” (rows 50-51).
“1.2. Interpersonal and intimate violence” (row 80).
“Relationships and directionality between violence, AOD use, and mental health conditions” (rows 121-122).
“Factors, the directionality of associations, and the social determinants yet to be explored” (rows 166-167).
- The introduction effectively justifies the need for the study by highlighting the gaps in the existing literature, such as the unclear directionality of the relationships between interpersonal violence, AOD use, and mental health conditions during the pandemic. Additionally, the inclusion of specific data from Mexico further strengthens the rationale for conducting the study in this context.
We include data from Mexico provided by PAHO to strengthen the rationale for conducting the study in this country (rows 65 - 67).
“In Mexico, PAHO (3) identified higher rates of YLD due to interpersonal and intimate violence between 2000 and 2019. It observed a rate of 64.73 per 100,000 Mexicans, 86.49 for women, and 45.65 for men in 2019.”
We are also including data from Mexico about how the experience of abuse is related to to strengthens the rationale for conducting the study in this country (137 -138).
“In Mexico, Morales et al. (22) reported experiences of emotional and physical abuse related to stress, sadness, and anxiety.”
We include data from Mexico about binge drinking and mental health symptoms to strengthens the rationale for conducting the study in this country (rows 162-164).
“In Mexico, binge drinking was related to stress, sadness, and anxiety during the first year of the pandemic (22).”
- The inclusion of studies conducted during the COVID-19 pandemic, as well as pre-pandemic research, helps establish a comprehensive understanding of the topic. However, it would be beneficial to briefly mention any limitations or gaps in the existing studies to further emphasize the need for the current research.
We include the limitation in the existing studies to emphasize the need for current research (rows 177 - 183).
“However, reports on the prevalence or incidence of violence, AOD, and mental health conditions have been based on separate studies conducted at the start of the pandemic or found in data obtained earlier without knowing what happened afterwards. Such studies have also suggested an unclear directionality of relationships between these harmful effects. Thus, describing the last prevalence and an integrated model of directionality regarding relationships would help slow the progression of social determinants, drug abuse, and mental health conditions.”
- The overall language and clarity of the introduction are good. However, there are a few sentences that could be rephrased for better readability and understanding. Additionally, proofreading for minor grammatical errors and inconsistencies would further improve the quality of the manuscript
We have rephrased some sentences and proofread them throughout the paper. This is an international team and as such we worked back and forth in two languages. We hope the revision is better.
Methods
- The methods section provides detailed information about the study design, participants, instruments, and procedures. Overall, the section is well-structured and provides sufficient information to understand the study methodology. However, there are a few areas where the description could be further improved for clarity and completeness.
The section starts with a clear description of the number of participants surveyed and their demographic characteristics. However, it would be helpful to include additional information about the recruitment process and any inclusion/exclusion criteria, if applicable. This would provide a better understanding of how the sample was obtained and ensure transparency in participant selection.
We have included additional information about the recruitment process and inclusion/exclusion criteria to a better understanding of how the sample was obtained and to ensure transparency in participant selection (rows 221 – 234).
“Participants were invited to enroll in the web-based application between September 1, 2021, and August 31, 2022, through conferences in the media to obtain a link available on the public Mexican Health Ministry website and the official institutional website of the leading public university in Mexico. Participants were asked to log into the system with their email to identify their participation. Inclusion criteria were being of legal age, having at least completed high school, and residing in Mexico. Exclusion criteria were being under 17 or over 25 not having completed high school or being a healthcare provider. We also considered the criteria for internet E-surveys, such as data protection, development, testing, contact mode, advertising the survey, mandatory/voluntary participation, completion rate, cookies used, IP check, log file analysis, registration, and atypical timestamp considerations (37). Since the technological system does not allow non response rates, 100% of the participants were volunteers who completed the questionnaire. Our sample was therefore not homogeneous.”
- The section briefly mentions the process of obtaining informed consent, but it would be beneficial to provide more details. Specifically, it would be helpful to explain how participants were informed about the purpose of the study, the voluntary nature of participation, and their rights regarding data confidentiality and withdrawal. Additionally, it would be useful to mention whether any ethical approval was obtained for the study and provide the name of the approving institution.
We added information about the purpose of the study and the voluntary nature of their participation in the informed consent section which includes data confidentially and withdrawal are explained (rows 262 - 267):
“Regarding informed consent, researchers informed participants that the purpose of the survey was to understand mental health risks and how to deal with them. We also told participants that confidentiality would be ensured since we calculated general averages. Participants were told that their participation was voluntary, that findings would be used for epidemiological research and that they could refuse to comply with data requests and drop out at any point in the study.”
The ethical approval for the study and the name of the institution that approved it are included (rows 277 - 279).
“The protocol was approved with code FPSI/422/CEIP/157/2020 by the Institutional Review Board of the Psychology Faculty Ethics Committee on Applied Research at the National Autonomous University of Mexico.”
- The section describes the instruments used in the study, namely the Life Events Checklist 5th Edition (LEC-5), WHO Alcohol, Smoking, and Substance Involvement Screening Test (ASSIST), Major-Depressive-Episode (MDE) checklist, Generalized Anxiety (GA) scale, and Posttraumatic Checklist (PCL-5). While the instruments are briefly explained, it would be beneficial to provide more information about the psychometric properties of these instruments, such as their reliability and validity coefficients. Additionally, it would be useful to mention the specific scales or subscales used within each instrument and provide a brief rationale for their selection.
We have added the reliability and validity coefficients of the scales and moved the information at the beginning of each instrument description to provide a brief rationale for their selection (rows 286 - 292).
“…divided into four scales/factors. Four items asked about victimizing interpersonal violence, four about victimizing intimate violence, four about perpetrating interpersonal violence, and two about perpetrating intimate violence, in the previous six months (see Appendix A]. Reliability values of the subscales fluctuated between an α = 0.68 and α = 0.76 (see Table 2). Confirmatory factor analysis (CFA) found a good test factor structure (e.g., [13] = 343.566, p<0.001, an RMSEA = 0.059, a CFI = 0.949 and a TLI = 0.917). Note that…”
We moved ASSIST psychometric property information (rows 307 - 311).
“Reliability values fluctuated between α = 0.80 for the alcohol dimension and α = 0.91 for stimulants (42; see Table 2). Confirmatory factor analysis (CFA) found a good test factor structure ( [1,583] = 50,863.65, p<0.001, an RMSEA = 0.040, an SRMR = 0.032, a CFI = 0.920 and a TLI = 0.913).”
We moved MDE psychometric data information (rows 334 - 337).
“The MDE has good validity and reliability coefficients (43). The α = 0.92 and Confirmatory factor analysis (CFA) was found to have a good checklist factor structure ( [32] = 2,643.99, p<0.001, an RMSEA = 0.067, an SRMR = 0.023, a CFI = 0.975, and a TLI = 0.965)”.
We moved GA psychometric data information (rows 354 - 356).
“The α = 0.94 and Confirmatory Factor Analysis (CFA) found a good scale factor structure ( [5] = 350.57, p<0.001, an RMSEA = 0.061, an SRMR = 0.007, an CFI = 0.996, and a TLI = 0.992)”.
We moved PCL-5 psychometric data information (rows 365 - 373).
“PCL-5 has shown good validity and reliability coefficients (43). The α = 0.96 and Confirmatory factor analysis (CFA) found a good checklist factor structure ( [161] = 5,648.34, p<0.001, an RMSEA = 0.077, an SRMR = 0.040, a CFI = 0.9375, and a TLI = 0.924). Blevins et al., (46) reported that the four-factor structure was a model with a good fit ( [164] = 558.18, p<.001, a CFI = 0.91, a TLI = 0.89, an RMSEA = 0.07, and an SRMR = 0.05; alpha = 0.94), whose optimal score of 31 (out of a total of 80) yielded a sensitivity of 0.77, a specificity of 0.96, an efficiency of 0.93 and a quality of efficiency of 0.73”.
We have mentioned that we used all subscales of the test and included the name scales (row 283).
“… using all scales”
(row 286).
“…four scales/factors…”
(row 301).
“…ten groups/scales…”
(row 332).
“…consists of one scale…”
(row 370).
“… four-factor structure…”
- The section describes the procedure followed in the study, including the invitation of participants and data collection period. However, it would be helpful to provide more information about how participants accessed the web-based application and whether any reminders were sent to increase response rates. Additionally, it would be beneficial to explain how the feedback and counseling were provided to participants and what resources were made available to them.
We have included information about how participants accessed the web-based application and about internet E-survey criteria that helps with completion rate and atypical timestamp considerations (224 - 234).
“Participants were asked to log into the system with their email to identify their participation. Inclusion criteria were being of legal age, having at least completed high school, and residing in Mexico. .. We also considered the criteria for internet E-surveys, such as data protection, development, testing, contact mode, advertising the survey, mandatory/voluntary participation, completion rate, cookies used, IP check, log file analysis, registration, and atypical timestamp considerations (37). Since the technological system does not allow non response rates, 100% of the participants were volunteers who completed the questionnaire. Our sample was therefore not homogeneous”.
We also clarified that we used written feedback by an algorithm and counseling by Zoom and the Zoiper switcher (rows 268 to 273).
“…immediate written feedback was provided through a programmed algorithm, including psychoeducational tools (such as infographics, videos, and Moodle ® courses on COVID-19, self-care, relaxation techniques, problem-solving, and socioemotional management skills). Phone numbers were provided to obtain remote psychological counseling by Zoom or phone through the Zoiper ® 3.5 switcher from the Health Ministry and public university services.”
- To enhance the study further, it would be beneficial to consider the potential moderating or mediating effects of other factors.
We clarified that we used the SEM analysis to assess moderating and mediating effects between our main factors to probe our hypothesis (rows 359 to 465).
“We analyzed the predictive models between variables, evaluating the mediating effects of using AOD between violence and mental health symptoms and the moderating effects of the role of sex and educational attainment, with the chi-square test and their fit indices through the SEM with a mixture of continuous and categorical variables (35). We represented the final models with good fit indices that proved our hypothesis.”
We also added recommended use of additional analysis to evaluate the moderating and mediating effects of other factors already suggested in the limitations section in rows 1015 to 1017:
“Considering these factors and their mediating or moderating effect between violence, AOD use, and mental health conditions could contribute to a better understanding of these socio-demographic and socially relevant worldwide conditions.”
Results
- The authors present the results in a clear and organized manner, making it easy to understand the key findings of the study. However, there are a few suggestions and points that need to be addressed to improve the clarity and presentation of the results.
The authors mention that the data fitting was adequate, with CFIs and TLIs > 0.90, RMSEAs < 0.08, and SRMRs < 0.06. It would be helpful if they provide a brief explanation of these fit indices for readers who may not be familiar with them.
We have included a sentence about fit indices in the introduction section, where they are mentioned for the first time (row 194).
“…, based on a fit function given a specific estimation method,”
We have also included a brief explanation of these fit indices in the same section (rows 199 to 203).
“The RMSEA and the SRMR are absolute indices determining the distance between a hypothesized and a perfect model. CFI and TLI are incremental fit indices that compare the fit of the hypothesized model with that of a baseline model (a model with the worst fit).”
We also included brief information about the way of comparing models with their baseline models in the data analysis section (rows 416 to 424).
“Specifically, we calculated the Model Optimization Method, the number of free parameters, observations, and missing patterns to validate the models. We used the Model Test User Model with Test statistics, degrees of freedom, p-value (chi-square), and the Model Test Baseline Model. Then we compared the User Model to the Baseline Model using the CFI and TLI. We calculated the RMSEA with a 90% confidence interval-lower-upper ≤ 0.05, and the SRMR, their Parameter Estimates with Standard Error and Hessian Observed Information to determine the distance between a hypothesized model and a perfect model.”
- In Table 2, where the results from the factor models are presented, it would be beneficial to provide more information about the specific factors or latent variables included in each scale. This would help readers understand the underlying constructs being measured by the different scales.
We have differentiated the scales names in each row of table 2 and added the same differentiation in the note of table 2 (rows 483 - 488).
Scales-factors |
X² |
df |
p≤ |
RMSEA |
Confident Interval |
SRMR |
CFI |
TLI |
Cronbach´s alpha |
LEC-5 |
|
|
|
|
|
|
|
|
|
Victimizing Interpersonal Violence |
343.566 |
13 |
0.001 |
0.059 |
0.053-0.064 |
|
0.949 |
0.917 |
0.76 |
Victimizing Intimate Violence |
6.087 |
2 |
0.048 |
0.017 |
0.001-0.032 |
|
0.999 |
0.997 |
0.76 |
Perpetrating Interpersonal Violence |
94.634 |
2 |
0.001 |
0.079 |
0.066-0.093 |
|
0.952 |
0.855 |
0.68 |
Perpetrating Intimate Violence |
0.000 |
0 |
0.000 |
0.000 |
0.000-0.000 |
|
1.000 |
1.000 |
0.68 |
ASSIST |
|
|
|
|
|
|
|
|
|
Once in Lifetime |
272.565 |
35 |
0.001 |
0.030 |
0.027-0.034 |
|
0.981 |
0.975 |
0.60 |
Tobacco |
86.199 |
4 |
0.001 |
0.053 |
0.043-0.063 |
0.013 |
0.994 |
0.986 |
0.83 |
Alcohol |
163.646 |
8 |
0.001 |
0.051 |
0.045-0.058 |
0.018 |
0.986 |
0.973 |
0.77 |
Cannabis |
155.431 |
7 |
0.001 |
0.053 |
0.046-0.061 |
0.013 |
0.992 |
0.982 |
0.85 |
Cocaine |
190.925 |
6 |
0.001 |
0.064 |
0.057-0.072 |
0.014 |
0.992 |
0.981 |
0.87 |
Stimulants |
161.657 |
6 |
0.001 |
0.059 |
0.051-0.067 |
0.008 |
0.995 |
0.988 |
0.92 |
Inhalants |
65.051 |
3 |
0.001 |
0.053 |
0.042-0.064 |
0.010 |
0.997 |
0.985 |
0.85 |
Sedatives |
94.670 |
7 |
0.001 |
0.041 |
0.034-0.049 |
0.010 |
0.995 |
0.990 |
0.86 |
Hallucinogens |
323.535 |
8 |
0.001 |
0.073 |
0.066-0.080 |
0.026 |
0.960 |
0.925 |
0.71 |
Opiods |
158.546 |
3 |
0.001 |
0.084 |
0.073-0.095 |
0.009 |
0.998 |
0.990 |
0.96 |
Others |
220.741 |
7 |
0.001 |
0.064 |
0.057-0.072 |
0.017 |
0.989 |
0.976 |
0.86 |
MDE |
|
|
|
|
|
|
|
|
|
Depression |
1,210.246 |
42 |
0.001 |
0.072 |
0.069-0.076 |
0.034 |
0.954 |
0.939 |
0.89 |
GA |
|
|
|
|
|
|
|
|
|
Anxiety |
74.940 |
5 |
0.001 |
0.043 |
0.035-0.052 |
0.005 |
0.998 |
0.995 |
0.93 |
PCL-5 |
|
|
|
|
|
|
|
|
|
Rexperimentation |
15.073 |
4 |
0.005 |
0.024 |
0.012-0.038 |
0.008 |
0.999 |
0.997 |
0.86 |
NACM |
229.317 |
10 |
0.001 |
0.068 |
0.061-0.076 |
0.020 |
0.983 |
0.964 |
0.86 |
Hyperarousal |
200.198 |
8 |
0.001 |
0.071 |
0.063-0.080 |
0.030 |
0.972 |
0.947 |
0.78 |
PTSD |
4535.593 |
162 |
0.001 |
0.076 |
0.074-0.078 |
0.044 |
0.916 |
0.901 |
0.93 |
Note. Test: LEC-5=Life Events Checklist, ASSIST= Alcohol, Smoking, and Substance Involvement Screening Test, MDE=Major Depressive Episode, GA= Generalized Anxiety, PCL-5= Post-traumatic Checklist. Scales: NACM= Negative Alterations in Cognition and Mood, PTSD= Post-traumatic Stress Disorder (including the two avoidance items). Categorical Variables do not show SRMR as Li (2021) recommended.
- The reliability range of the scales is mentioned, but it would be valuable to include more details about the specific reliability measures used (e.g., Cronbach's alpha) and their interpretation in the context of the study. Additionally, it would be helpful to provide a brief discussion of the implications of the reliability estimates for the interpretation of the scale scores.
We have included information about the interpretation of the Cronbach Alpha to provide a brief discussion of the implications of the reliability estimates for the interpretation of the scale scores in the context of the study in the data analysis section (rows 427 - 432).
“We also obtained a Cronbach’s Alpha test for each scale to determine the reliability of the dimensions. Cronbach´s alpha test is based on the item correlation in each scale represented by a number between 0 and 1. A value near to one means that items in the scales are more correlated with each other, evaluating the same latent variable and being stable over time. Cronbach makes it possible to determine the degree of independence between dimensions.”
We also included an interpretation sentence of the Cronbach alpha obtained in results section (rows 478 - 481).
“Items within each scale were therefore well correlated, evaluating latent variables consistently over time. Cronbach alpha makes it possible to determine the degree of independence between dimensions.”
We also included the interpretation of Cronbach alpha in the discussion section (rows 637 - 639).
“Findings suggest a good, stable structure of violence, harmful AOD use, depression, anxiety, and PTSD measurements. Latent variables were independent of each other.”
- Table 3 provides information about the distribution of participants at risk for various outcomes, such as violence, harmful AOD use, depression, anxiety, and PTSD symptoms. While the table is informative, it would be beneficial to include additional information on the sample size and the specific criteria used to define "at-risk" status for each outcome. This would help readers understand the magnitude and clinical significance of the reported percentages.
We have included the sample size in table 3 and specified the criteria used to define “at-risk” status in the note for each outcome to help understand the magnitude and clinical significance of the reported percentages instrument section and in table 3 (rows 518 to 528).
“
TOTAL |
|||||||||||||||
n |
% |
||||||||||||||
7420 |
100 |
||||||||||||||
Victimizing Interpersonal Violence |
Victimizing Intimate Violence |
||||||||||||||
n |
% |
n |
% |
||||||||||||
1853 |
25.00 |
1874 |
25.26 |
||||||||||||
Men |
Women |
High school |
University |
Men |
Women * |
High school |
University |
||||||||
n |
% |
n |
% |
n |
% |
n |
% |
n |
% |
n |
% |
n |
% |
n |
% |
579 |
25.02 |
1274 |
24.95 |
407 |
24.10 |
1446 |
25.23 |
515 |
22.26 |
1359 |
26.62 |
427 |
25.28 |
1447 |
25.25 |
Perpetrating Interpersonal Violence |
Perpetrating Intimate Violence |
||||||||||||||
n |
% |
n |
% |
||||||||||||
1742 |
23.48 |
1141 |
15.38 |
||||||||||||
Men |
Women |
High school |
University |
Men |
Women* |
High school |
University |
||||||||
n |
% |
n |
% |
n |
% |
n |
% |
n |
% |
n |
% |
n |
% |
n |
% |
532 |
22.99 |
1210 |
23.70 |
384 |
22.73 |
1358 |
23.70 |
315 |
13.61 |
826 |
16.18 |
253 |
14.98 |
888 |
15.49 |
Harmful tobacco use |
Harmful alcohol use |
||||||||||||||
n |
% |
n |
% |
||||||||||||
1919 |
25.90 |
1497 |
20.20 |
||||||||||||
Men * |
Women |
High school * |
University |
Men * |
Women |
High school * |
University |
||||||||
n |
% |
n |
% |
n |
% |
n |
% |
n |
% |
n |
% |
n |
% |
n |
% |
683 |
29.50 |
1236 |
24.20 |
498 |
29.50 |
1421 |
24.80 |
547 |
23.60 |
950 |
18.60 |
371 |
22.00 |
1126 |
19.60 |
Harmful cannabis use |
Harmful cocaine use |
||||||||||||||
n |
% |
n |
% |
||||||||||||
927 |
12.50 |
137 |
1.80 |
||||||||||||
Men * |
Women |
High school * |
University |
Men * |
Women |
High school * |
University |
||||||||
n |
% |
n |
% |
n |
% |
n |
% |
n |
% |
n |
% |
n |
% |
n |
% |
365 |
15.80 |
562 |
11.00 |
239 |
14.20 |
688 |
12.00 |
62 |
2.70 |
75 |
1.50 |
48 |
2.80 |
89 |
1.60 |
Harmful stimulants use |
Harmful inhalants use |
||||||||||||||
n |
Total % |
n |
% |
||||||||||||
57 |
0.80 |
22 |
0.30 |
||||||||||||
Men * |
Women |
High school * |
University |
Men |
Women |
High school |
University |
||||||||
n |
% |
n |
% |
n |
% |
n |
% |
n |
% |
n |
% |
n |
% |
n |
% |
26 |
1.10 |
31 |
0.60 |
22 |
1.30 |
35 |
0.60 |
10 |
0.40 |
12 |
0.20 |
8 |
0.50 |
14 |
0.20 |
Harmful sedatives use |
Harmful hallucinogens use |
||||||||||||||
n |
% |
n |
% |
||||||||||||
357 |
4.80 |
203 |
2.70 |
||||||||||||
Men |
Women * |
High school |
University |
Men * |
Women |
High school |
University |
||||||||
n |
% |
n |
% |
n |
% |
n |
% |
n |
% |
n |
% |
n |
% |
n |
% |
81 |
3.50 |
276 |
5.40 |
84 |
5.00 |
273 |
4.80 |
85 |
3.70 |
118 |
2.30 |
56 |
3.30 |
147 |
2.60 |
Harmful opioids use |
Harmful use of other drugs |
||||||||||||||
n |
% |
n |
% |
||||||||||||
8 |
0.10 |
100 |
1.30 |
||||||||||||
Men |
Women |
High school |
University |
Men |
Women |
High school |
University |
||||||||
n |
% |
n |
% |
n |
% |
n |
% |
n |
% |
n |
% |
n |
% |
n |
% |
3 |
0.10 |
5 |
0.10 |
2 |
0.10 |
6 |
0.10 |
32 |
1.40 |
68 |
1.30 |
25 |
1.50 |
75 |
1.30 |
Poly Drug Use |
|
||||||||||||||
n |
% |
|
|||||||||||||
1405 |
18.93 |
||||||||||||||
Men * |
Women |
High school |
University |
||||||||||||
n |
% |
n |
% |
n |
% |
n |
% |
||||||||
525 |
22.69 |
880 |
17.23 |
352 |
20.84 |
1053 |
18.37 |
||||||||
Depression |
Generalized Anxiety |
||||||||||||||
n |
% |
n |
% |
||||||||||||
3299 |
44.46 |
3553 |
47.90 |
||||||||||||
Men |
Women * |
High school * |
University |
Men |
Women * |
High school |
University |
||||||||
n |
% |
n |
% |
n |
% |
n |
% |
n |
% |
n |
% |
n |
% |
n |
% |
868 |
37.51 |
2431 |
47.61 |
794 |
47.01 |
2505 |
43.71 |
979 |
42.30 |
2574 |
50.40 |
834 |
49.40 |
2719 |
47.40 |
PTSD symptoms |
Comorbidity |
||||||||||||||
n |
% |
n |
% |
||||||||||||
2187 |
29.47 |
2713 |
36.56 |
||||||||||||
Men |
Women * |
High school * |
University |
Men |
Women |
High school * |
University |
||||||||
n |
% |
n |
% |
n |
% |
n |
% |
n |
% |
n |
% |
n |
% |
n |
% |
565 |
24.42 |
1622 |
31.77 |
566 |
33.51 |
1621 |
28.28 |
708 |
30.60 |
2005 |
39.27 |
685 |
40.56 |
2028 |
35.39 |
Note. * significant differences between groups < 0.05 according to the Chi-square analysis. At-risk groups were included in the instruments section. This means: 1) a score over one for victimizing perpetrating, interpersonal or intimate violence; 2) a score over four for all drugs except alcohol, for which the score has to be over 11 to meet the criteria; 3) more than one harmful AOD use for polydrug use; 4) Meeting criteria A and B for depression; 5) an average of over 60% for anxiety; 6) A two -response option or more for at least one of the B-items, one of the C-items, two of the D-items, two of the E-items, and bothersome symptoms for over a month for PTSD; and 7) more than one group of mental health symptoms for comorbidity.“
- The authors discuss significant differences between groups (e.g., men vs. women, high school vs. university graduates) in terms of the proportions at risk for different outcomes. However, it is not clear how these differences were assessed statistically. Providing information about the statistical tests used and the significance levels would enhance the rigor and transparency of the analysis.
We have included the information about how differences between groups were assessed statistically to enhance the rigor and transparency of the analysis, in the data analysis section (rows 438 - 443).
“We compared the distribution of the participants by risk level by sex and educational attainment. We therefore performed chi-square tests, considering p values under 0.05, on participants’ distribution, by groups of risk from violence (victimizing interpersonal and intimate violence, perpetrating interpersonal and intimate violence), harmful AOD use, depression, anxiety, and PTSD symptoms, and by sex or educational attainment of the sample.”
And in table 3 (rows 519 - 520).
“Note. * significant differences between groups < 0.05 according with the Chi-square analysis.”
- Lastly, the authors mention that all path coefficients in the models were significant at p < 0.01 or less. While this information is important, it would be helpful to include the actual values of the path coefficients to provide a quantitative understanding of the relationships between variables.
We have clarified that the significant p<0.01 values refer to the path coefficients named at the beginning of the paragraph, adding an example, (rows 591 - 592).
“All these path coefficients were significant at p < 0.01 or less (e.g., bTob = 0.190, from victimizing intimate violence to harmful use of tobacco).”

Reviewer 2 Report
1. The abstract is scholarly written. It follows the IMRAD format. However, the word count needs to be slashed down a bit for conciseness. Probably, the introductory section could be condensed to only the main drivers of the purpose of the study.
2. The introduction offers a resounding theoretical foundation for the study. Enough background information from previous studies to ground the study. It intelligently exposes the gaps in previous studies and how this current study fits into the broader picture of the field. It highlights the need for the study in improving public policies and preventive programs that are cost effective and viable in arresting mental health disorders and its allied antecedents such as depression, substance use, stress and anxiety in relation to interpersonal and intimate violence among Mexican youth during the Covid-19 period.
3. The method section is academically rigorous and offers the required comprehensive information that would ensure the replication of the study. The instruments, recruitment of study participants, procedure and analysis are scholarly discussed. I was wondering if pre-testing of the instruments across a small section of the same population was done to affirm the already validated instruments in the Mexican context before the field administration. Were other validation procedures used? If yes, these must be described to further vouch the validity of the tools/instruments for data collection, especially in the Mexican context.
4. The results have been excellent presented and discussed. Graphical tools have been used in illuminating the data garnered in statistical formats and they are quite objective. In the discussions, the findings have been intelligently compared with previous studies. The authors as required of scientific writers, discusses the possible differences in their findings and those earlier reported in a more academic manner that merits commendation. It's flawless.
5. I am worried that the concluding section is too loaded. It looks as if the authors have re-looped some of the significant findings to the section, though it's evident that some interesting revelations have also been added. I suggest these should be moved to the discussion section. The section must highlight the significant findings and what they mean to the field of mental health generally and more specifically, sound tentative conclusions that could be drawn from them. These, though present, have been entangled in other very detailed discussions not meant to be in that section.
6. The limitations and further areas of research explicitly presented demonstrates the authors sensitiveness to the research process as a continuum process or cycle and modestly acknowledge the possible limitations in the work. This is excellent.
Author Response
We really appreciate every word from you and the valuable comments. Here is our attendance of them. Thank you very much.
Corrections cover letter
Reviewer 2
- The abstract is scholarly written. It follows the IMRAD format. However, the word count needs to be slashed down a bit for conciseness. Probably, the introductory section could be condensed to only the main drivers of the purpose of the study.
We have reduced the abstract for conciseness (rows 11 - 35).
“The COVID-19 pandemic may have increased interpersonal and intimate violence, harmful use of alcohol and other drugs (AOD) , and mental health problems. This study uses a valid path model to describe relationships between these conditions of young Mexicans during the second year of the pandemic. A sample of 7,420 Mexicans ages 18 to 24—two-thirds of whom are women—completed the Life Events Checklist, the Alcohol, Smoking, and Substance Involvement Screening Test, the Major Depressive Episode Checklist, the Generalized Anxiety Scale, and the Post-traumatic Stress Disorder (PTSD) Checklist. Young Mexicans reported higher rates of victimization and perpetration of interpersonal and intimate violence and mental health symptomatology than those noted pre- and in the first year of the pandemic. Harmful use of AOD rates were similar to those reported by adolescents before. Findings suggested asymmetric victimization and perpetration of intimate violence by gender (with women at a higher risk). More men than women have engaged in the harmful use of AOD (except for sedatives, which more women abuse). More women than men were at risk of all mental health conditions. The path model indicates that being a victim of intimate violence predicts harmful use of tobacco, alcohol, cocaine, and sedatives, depression, anxiety, and specific PTSD symptoms (such as re-experimentation and avoidance symptoms). Being a victim of interpersonal violence resulted in severe PTSD symptoms (including avoidance, negative alterations in cognition-mood, and hyperarousal signs). Harmful use of sedatives predicted depressive symptoms. Men´s victimizing intimate violence model contrasted with that of women, which included being the victim of interpersonal violence and severe PTSD symptoms. The high school youth model had three paths -victimizing intimate violence, victimizing-interpersonal abuse, and sedative use, which predicted depression. Our findings could serve as the basis for future studies exploring mechanisms that predict violence to develop cost-effective preventive programs and public policies and to address mental health conditions during community emergencies.”
About the introduction, we have included subheadings to improve readability and facilitate navigation through the text (rows 50 - 51).
“1.1. Violence, AOD use, and mental health conditions before and in the first year of the pandemic”;
(row 80).
“1.2. Interpersonal and intimate violence”;
(rows 121 – 122).
“Relationships and directionality between violence, AOD use, and mental health conditions”;
(rows 166 - 167).
“Factors, the directionality of associations, and the social determinants yet to be explored”
- The method section is academically rigorous and offers the required comprehensive information that would ensure the replication of the study. The instruments, recruitment of study participants, procedure and analysis are scholarly discussed. I was wondering if pre-testing of the instruments across a small section of the same population was done to affirm the already validated instruments in the Mexican context before the field administration. Were other validation procedures used? If yes, these must be described to further vouch the validity of the tools/instruments for data collection, especially in the Mexican context.
We have added the reliability and validity coefficients of the scales and moved the information at the beginning of each instrument description to provide a brief rationale for their selection (rows 286 - 292).
“…divided into four scales/factors. Four items asked about victimizing interpersonal violence, four about victimizing intimate violence, four about perpetrating interpersonal violence, and two about perpetrating intimate violence, in the previous six months (see Appendix A]. Reliability values of the subscales fluctuated between an α = 0.68 and α = 0.76 (see Table 2). Confirmatory factor analysis (CFA) found a good test factor structure (e.g., [13] = 343.566, p<0.001, an RMSEA = 0.059, a CFI = 0.949 and a TLI = 0.917). Note that…”
We moved ASSIST psychometric property information (rows 307 - 311).
“Reliability values fluctuated between α = 0.80 for the alcohol dimension and α = 0.91 for stimulants (42; see Table 2). Confirmatory factor analysis (CFA) found a good test factor structure ( [1,583] = 50,863.65, p<0.001, an RMSEA = 0.040, an SRMR = 0.032, a CFI = 0.920 and a TLI = 0.913).”
We moved MDE psychometric data information (rows 334 - 337).
The MDE has good validity and reliability coefficients (43). The α = 0.92 and Confirmatory factor analysis (CFA) was found to have a good checklist factor structure ( [32] = 2,643.99, p<0.001, an RMSEA = 0.067, an SRMR = 0.023, a CFI = 0.975, and a TLI = 0.965)”.
We moved GA psychometric data information (rows 354 to 356).
“The α = 0.94 and Confirmatory Factor Analysis (CFA) found a good scale factor structure ( [5] = 350.57, p<0.001, an RMSEA = 0.061, an SRMR = 0.007, an CFI = 0.996, and a TLI = 0.992)”.
We moved PCL-5 psychometric data information (rows 365 - 373).
“PCL-5 has shown good validity and reliability coefficients (43). The α = 0.96 and Confirmatory factor analysis (CFA) found a good checklist factor structure ( [161] = 5,648.34, p<0.001, an RMSEA = 0.077, an SRMR = 0.040, a CFI = 0.9375, and a TLI = 0.924). Blevins et al., (46) reported that the four-factor structure was a model with a good fit ( [164] = 558.18, p<.001, a CFI = 0.91, a TLI = 0.89, an RMSEA = 0.07, and an SRMR = 0.05; alpha = 0.94), whose optimal score of 31 (out of a total of 80) yielded a sensitivity of 0.77, a specificity of 0.96, an efficiency of 0.93 and a quality of efficiency of 0.73”.
- I am worried that the concluding section is too loaded. It looks as if the authors have re-looped some of the significant findings to the section, though it's evident that some interesting revelations have also been added. I suggest these should be moved to the discussion section. The section must highlight the significant findings and what they mean to the field of mental health generally and more specifically, sound tentative conclusions that could be drawn from them. These, though present, have been entangled in other very detailed discussions not meant to be in that section.
We moved the reflections about significant findings to the discussion section, letting the conclusion sections highlight the main findings to underline what they meant to the mental health field. Changes were made (rows 724 - 733).
“The proportion of young people suffering or perpetrating violence, using AOD, and having mental health problems can be explained by conditions during the pandemic. Glowacz et al. (17) and Kourti et al. (11) have suggested that lockdown or losses during the pandemic could explain these circumstances. Future research, however, could explore how sociodemographic settings related to the pandemic, such as social distance, loss of loved ones and losing jobs were related to violence, drug use, and mental health illness in the second year of the pandemic. Meanwhile, validating the structure of the variables paved the way for path analysis and proposals for future research. Describing the directionality of links between the variables could contribute to future research and prevent certain conditions in future pandemics.”
(rows 793 - 804).
“One hypothesis about the complexity of PTSD is related to forms of violence. Keely et al. (50) have suggested that complex PTSD can be described as pervasive problems with affect regulation (NACM), persistent beliefs about oneself as diminished (succumbing to adverse circumstances), persistent difficulties in sustaining relationships (feeling close to others), and disturbances causing significantly impaired functioning. These severe symptoms were reported by young Mexican women when they experienced interpersonal violence rather than intimate abuse. Victimizing intimate violence was related to normal PTSD symptoms in addition to depression, anxiety, and drug use -as a possible means of coping. Future longitudinal research could analyze complex PTSD related to forms of violence. It is essential, however, to study this relationship in a context where coping skills could halt the progression of acute stress symptoms to complex PTSD.”
(rows 809 - 811).
“Additional research should confirm whether a violence-escalating mechanism can occur once young people have been interpersonally or intimately victimized (16).”
(rows 814 – 826).
“Scott-Storey and collaborators (16) have already proposed that asymmetries in victimizing intimate violence by sex may result from differences in the perception of violence by sex in a context of social inequities, and normalized violation of human rights as happens in Mexico (51). Both men and women suffer intimate violence. However, forms of violence and their consequences may differ by sex due to the patriarchal culture and the role of power and control in societies. Although men disclose the forms of violence suffered, they seem to view emotional and sexual abuse as more dangerous than physical violence. Women can endure several forms of violence for long periods of time, suffering greater consequences (51). Men and women, however, seem to cope with victimizing intimate violence through AOD use (52). Future longitudinal research should therefore address forms of violence, gender interaction, and the consequences of perceived abuse by sex and culture in several low-income countries where human rights are routinely violated.”
(rows 835 - 844).
“Findings constitute a baseline to explore the hypothesis of the mechanisms behind these paths. Hernandez (30), Gubi et al. (31), and Dos-Santos et al. (25) suggest that lower educational attainment predicts violence, while Craig et al. (19) have reported that age of onset of drug use is lower in adolescents with lower educational attainment. However, the mechanisms in the paths of the model for participants with lower educational attainment could be addressed in future longitudinal studies. Another hypothesis concerns the role of drug use as a self-medicating mechanism. Future longitudinal research could determine whether the use of tobacco, alcohol, cocaine, and sedatives is a coping mechanism to numb feelings related to violence or to avoid thinking about experiencing violence.”
(rows 846 - 854).
“In general, participants showed no relationship between AOD use and perpetrating intimate or interpersonal violence. Our Mexican sample may be too young to show that drug use predicts the perpetration of violence as Ismayilova (53) has predicted with a sample of older participants. The association between drug use and perpetrating interpersonal and intimate violence may be linked to being older, several life conditions, being a caregiver, having lower educational attainment, or experiencing certain socioeconomic conditions. Future research could therefore explore these conditions that could explain the associations between AOD use and violence perpetrated as Islam et al. (54) proposed.”

Reviewer 3 Report
The authors conducted a very important study in which an attempt was made to identify the relationship between the use AOD, violence in the intimate sphere and depression, anxiety, and stress during the COVID-19 Pandemic. It should be noted that a very extensive work has been carried out involving a large amount of data and their analysis. The authors used modern methods of analysis and the reliability of the results is beyond doubt. At the same time, it would be fair to interpret the results taking into account the cultural characteristics of the persons involved in the study, and in the discussion it is also necessary to take into account that the results of other authors are often obtained on a culturally different sample. Another important point that needs to be taken into account is the primary-secondary nature of the behavior factor. In combined symptoms, as a rule, mental disorders are primary in relation to behavioral ones. Therefore, the use of AOD may be secondary to mental disorders, but a provoking factor of violence. In addition, and the study shows this, various types of surfactants can be used as a result of human victimization. The article needs a clearer differentiation of the results by type of AOD, by role in violence, by gender, by age/type of educational organization.
Author Response
We really appreciate every word and valuable comment. Here, it is our attendance to them. Thank you very much.
Corrections cover letter
Reviewer 3
- At the same time, it would be fair to interpret the results considering the cultural characteristics of the persons involved in the study, and in the discussion, it is also necessary to consider that the results of other authors are often obtained on a culturally different sample.
We included the sentences about the cultural and socio-economical context in Mexico in the discussion section, (rows 814 - 819).
“Scott-Storey and collaborators (16) have already proposed that asymmetries in victimizing intimate violence by sex may result from differences in the perception of violence by sex in a context of social inequities, and normalized violation of human rights as happens in Mexico (51). Both men and women suffer intimate violence. However, forms of violence and their consequences may differ by sex due to the patriarchal culture and the role of power and control in societies.”
(rows 858 - 860).
“A valid measurement of variables has suggested that Mexican youth reported higher levels of violence than those reported before the pandemic, but in a culturally different sample from a low-middle income country (LMIC).”
(row 881).
"… predictive patterns in LMIC.”
- Another important point that needs to be taken into account is the primary-secondary nature of the behavior factor. In combined symptoms, as a rule, mental disorders are primary in relation to behavioral ones. Therefore, the use of AOD may be secondary to mental disorders, but a provoking factor of violence. In addition, and the study shows this, various types of surfactants can be used as a result of human victimization. The article needs a clearer differentiation of the results by type of AOD, by role in violence, by gender, by age/type of educational organization.
We have clarified the role of directionality of the variables and needed studies about it in the discussion section (rows 846 - 854).
“In general, participants showed no relationship between AOD use and perpetrating intimate or interpersonal violence. Our Mexican sample may be too young to show that drug use predicts the perpetration of violence as Ismayilova (53) has predicted with a sample of older participants. The association between drug use and perpetrating interpersonal and intimate violence may be linked to being older, several life conditions, being a caregiver, having lower educational attainment, or experiencing certain socioeconomic conditions. Future research could therefore explore these conditions that could explain the associations between AOD use and violence perpetrated as Islam et al. (54) proposed.”
Reviewer 4 Report
Most of my comments relate to precision in the presentation of results and text editing issues.
The final paragraph of the introduction about the fitting quality of pathway models is not necessary. It is more noteworthy that such models show complex dependency mechanisms and a range of direct and indirect relationships.
A clear purpose and possibly research questions or hypotheses should be at the end of the introduction.
Unclear abbreviations Ha1 to Ha8 in lines 179+.
Neither in the methods nor in the description of the results is it stated how the OR was estimated.
Table 1 should be reworded so that the type of school is in the first column. Figure 3 is unreadable for the same reason.
Table 2 could be in an appendix below the main text.
Figure 1 is unreadable. It should be explained more clearly what the dependent variable was. If there are six dependent variables in Figure 1, it would be clearer to rearrange six separate readable figures.
Figure 2 would be easier to perceive in a landscape orientation.
Also, not very readable is Figure 3 with the models for the four groups. Here, it was more appropriate to rearrange the results in a table.
The notation of the literature is not in accordance with the rules given in the instructions to the authors.
In citing literature, a comma not a semicolon should be the separator.
Author Response
We really appreciate every word and the valuable suggestions. Here, it is our corrections. Thank you very much.
Corrections cover letter
Reviewer 4
- The final paragraph of the introduction about the fitting quality of pathway models is not necessary. It is more noteworthy that such models show complex dependency mechanisms and a range of direct and indirect relationships.
Thank you for the valuable comment. However, we have decided to maintain the fitting quality of the pathway because the first reviewer even recommended providing a brief explanation of these fit indices for readers who may need to become more familiar with them (rows 199 - 203).
“The RMSEA and the SRMR are absolute indices determining the distance between a hypothesized and a perfect model. CFI and TLI are incremental fit indices that compare the fit of the hypothesized model with that of a baseline model (a model with the worst fit).”
- A clear purpose and possibly research questions or hypotheses should be at the end of the introduction.
We have clarified the purpose of the study and the hypothesis, already at the end of the introduction section, (rows 210 - 218).
“We have several hypotheses (Ha), the main one being that victimizing interpersonal and intimate violence are associated with harmful AOD use and mental health symptoms, moderated by age and education demographics. The study therefore aims to explore whether victimizing interpersonal and intimate violence predicts harmful psychoactive substance use (Ha1), depression (Ha2), anxiety (Ha3), and PTSD symptoms (Ha4), with differences between sex and educational attainment in the context of the pandemic. In addition, it explores whether harmful AOD affects the perpetration of interpersonal and intimate violence (Ha5), depression (Ha6), anxiety (Ha7), and PTSD symptomatology (Ha8).”
- Unclear abbreviations Ha1 to Ha8 in lines 179+
We have clarified that Ha means Hypotheses (row 210).
“We have several hypotheses (Ha),…”
- Neither in the methods nor in the description of the results is it stated how the OR was estimated.
We have included the description of how we estimated the OR in the data analysis section (rows 450 - 456).
“Odd ratios were calculated through two variables (violence, AOD) with two categories each (at risk-not risk) and analyzing their contingency table, based on the Chi-square test and p-value. Odds ratios have values between 0 and positive infinity. Values around one indicate that one variable poses no risk to the other one. Values over one indicate the likelihood of one variable having risk over the other, while values under one indicate that the variable represents protection against the other variable.”
And in the results section (rows 531 - 533).
“Remember that values around one indicate that exposure to one variable poses no risk to the other one. Values over one indicate that exposure to one variable affects the other.”
- Table 1 should be reworded so that the type of school is in the first column. Figure 3 is unreadable for the same reason.
We have reworded Table 1 and Table 3 for better reading, also considering reviewer one comments; however, we decided to leave the sex in the first column because we describe results and discuss findings in this order: sex and attainment (rows 253-254).
|
|||||||
TOTAL |
|||||||
n |
% |
||||||
7420 |
100 |
||||||
Men |
Women |
High school |
University |
||||
n |
% |
n |
% |
n |
% |
n |
% |
2314 |
31.20 |
5106 |
68.8 |
1689 |
22.80 |
5731 |
77.20 |
And in rows 517 to 528:
“
TOTAL |
|||||||||||||||
n |
% |
||||||||||||||
7420 |
100 |
||||||||||||||
Victimizing Interpersonal Violence |
Victimizing Intimate Violence |
||||||||||||||
n |
% |
n |
% |
||||||||||||
1853 |
25.00 |
1874 |
25.26 |
||||||||||||
Men |
Women |
High school |
University |
Men |
Women * |
High school |
University |
||||||||
n |
% |
n |
% |
n |
% |
n |
% |
n |
% |
n |
% |
n |
% |
n |
% |
579 |
25.02 |
1274 |
24.95 |
407 |
24.10 |
1446 |
25.23 |
515 |
22.26 |
1359 |
26.62 |
427 |
25.28 |
1447 |
25.25 |
Perpetrating Interpersonal Violence |
Perpetrating Intimate Violence |
||||||||||||||
n |
% |
n |
% |
||||||||||||
1742 |
23.48 |
1141 |
15.38 |
||||||||||||
Men |
Women |
High school |
University |
Men |
Women* |
High school |
University |
||||||||
n |
% |
n |
% |
n |
% |
n |
% |
n |
% |
n |
% |
n |
% |
n |
% |
532 |
22.99 |
1210 |
23.70 |
384 |
22.73 |
1358 |
23.70 |
315 |
13.61 |
826 |
16.18 |
253 |
14.98 |
888 |
15.49 |
Harmful tobacco use |
Harmful alcohol use |
||||||||||||||
n |
% |
n |
% |
||||||||||||
1919 |
25.90 |
1497 |
20.20 |
||||||||||||
Men * |
Women |
High school * |
University |
Men * |
Women |
High school * |
University |
||||||||
n |
% |
n |
% |
n |
% |
n |
% |
n |
% |
n |
% |
n |
% |
n |
% |
683 |
29.50 |
1236 |
24.20 |
498 |
29.50 |
1421 |
24.80 |
547 |
23.60 |
950 |
18.60 |
371 |
22.00 |
1126 |
19.60 |
Harmful cannabis use |
Harmful cocaine use |
||||||||||||||
n |
% |
n |
% |
||||||||||||
927 |
12.50 |
137 |
1.80 |
||||||||||||
Men * |
Women |
High school * |
University |
Men * |
Women |
High school * |
University |
||||||||
n |
% |
n |
% |
n |
% |
n |
% |
n |
% |
n |
% |
n |
% |
n |
% |
365 |
15.80 |
562 |
11.00 |
239 |
14.20 |
688 |
12.00 |
62 |
2.70 |
75 |
1.50 |
48 |
2.80 |
89 |
1.60 |
Harmful stimulants use |
Harmful inhalants use |
||||||||||||||
n |
Total % |
n |
% |
||||||||||||
57 |
0.80 |
22 |
0.30 |
||||||||||||
Men * |
Women |
High school * |
University |
Men |
Women |
High school |
University |
||||||||
n |
% |
n |
% |
n |
% |
n |
% |
n |
% |
n |
% |
n |
% |
n |
% |
26 |
1.10 |
31 |
0.60 |
22 |
1.30 |
35 |
0.60 |
10 |
0.40 |
12 |
0.20 |
8 |
0.50 |
14 |
0.20 |
Harmful sedatives use |
Harmful hallucinogens use |
||||||||||||||
n |
% |
n |
% |
||||||||||||
357 |
4.80 |
203 |
2.70 |
||||||||||||
Men |
Women * |
High school |
University |
Men * |
Women |
High school |
University |
||||||||
n |
% |
n |
% |
n |
% |
n |
% |
n |
% |
n |
% |
n |
% |
n |
% |
81 |
3.50 |
276 |
5.40 |
84 |
5.00 |
273 |
4.80 |
85 |
3.70 |
118 |
2.30 |
56 |
3.30 |
147 |
2.60 |
Harmful opioids use |
Harmful use of other drugs |
||||||||||||||
n |
% |
n |
% |
||||||||||||
8 |
0.10 |
100 |
1.30 |
||||||||||||
Men |
Women |
High school |
University |
Men |
Women |
High school |
University |
||||||||
n |
% |
n |
% |
n |
% |
n |
% |
n |
% |
n |
% |
n |
% |
n |
% |
3 |
0.10 |
5 |
0.10 |
2 |
0.10 |
6 |
0.10 |
32 |
1.40 |
68 |
1.30 |
25 |
1.50 |
75 |
1.30 |
Poly Drug Use |
|
||||||||||||||
n |
% |
|
|||||||||||||
1405 |
18.93 |
||||||||||||||
Men * |
Women |
High school |
University |
||||||||||||
n |
% |
n |
% |
n |
% |
n |
% |
||||||||
525 |
22.69 |
880 |
17.23 |
352 |
20.84 |
1053 |
18.37 |
||||||||
Depression |
Generalized Anxiety |
||||||||||||||
n |
% |
n |
% |
||||||||||||
3299 |
44.46 |
3553 |
47.90 |
||||||||||||
Men |
Women * |
High school * |
University |
Men |
Women * |
High school |
University |
||||||||
n |
% |
n |
% |
n |
% |
n |
% |
n |
% |
n |
% |
n |
% |
n |
% |
868 |
37.51 |
2431 |
47.61 |
794 |
47.01 |
2505 |
43.71 |
979 |
42.30 |
2574 |
50.40 |
834 |
49.40 |
2719 |
47.40 |
PTSD symptoms |
Comorbidity |
||||||||||||||
n |
% |
n |
% |
||||||||||||
2187 |
29.47 |
2713 |
36.56 |
||||||||||||
Men |
Women * |
High school * |
University |
Men |
Women |
High school * |
University |
||||||||
n |
% |
n |
% |
n |
% |
n |
% |
n |
% |
n |
% |
n |
% |
n |
% |
565 |
24.42 |
1622 |
31.77 |
566 |
33.51 |
1621 |
28.28 |
708 |
30.60 |
2005 |
39.27 |
685 |
40.56 |
2028 |
35.39 |
Note. * significant differences between groups < 0.05 according to the Chi-square analysis. At-risk groups were included in the instruments section. This means: 1) a score over one for victimizing perpetrating, interpersonal or intimate violence; 2) a score over four for all drugs except alcohol, for which the score has to be over 11 to meet the criteria; 3) more than one harmful AOD use for polydrug use; 4) Meeting criteria A and B for depression; 5) an average of over 60% for anxiety; 6) A two -response option or more for at least one of the B-items, one of the C-items, two of the D-items, two of the E-items, and bothersome symptoms for over a month for PTSD; and 7) more than one group of mental health symptoms for comorbidity.“
- Table 2 could be in an appendix below the main text.
We have reworded Table 2 for a better reading; however, Authors decided to leave it as Table, under the main text (rows 483 - 488).
“
Scales-factors |
X² |
df |
p≤ |
RMSEA |
Confident Interval |
SRMR |
CFI |
TLI |
Cronbach´s alpha |
LEC-5 |
|
|
|
|
|
|
|
|
|
Victimizing Interpersonal Violence |
343.566 |
13 |
0.001 |
0.059 |
0.053-0.064 |
|
0.949 |
0.917 |
0.76 |
Victimizing Intimate Violence |
6.087 |
2 |
0.048 |
0.017 |
0.001-0.032 |
|
0.999 |
0.997 |
0.76 |
Perpetrating Interpersonal Violence |
94.634 |
2 |
0.001 |
0.079 |
0.066-0.093 |
|
0.952 |
0.855 |
0.68 |
Perpetrating Intimate Violence |
0.000 |
0 |
0.000 |
0.000 |
0.000-0.000 |
|
1.000 |
1.000 |
0.68 |
ASSIST |
|
|
|
|
|
|
|
|
|
Once in Lifetime |
272.565 |
35 |
0.001 |
0.030 |
0.027-0.034 |
|
0.981 |
0.975 |
0.60 |
Tobacco |
86.199 |
4 |
0.001 |
0.053 |
0.043-0.063 |
0.013 |
0.994 |
0.986 |
0.83 |
Alcohol |
163.646 |
8 |
0.001 |
0.051 |
0.045-0.058 |
0.018 |
0.986 |
0.973 |
0.77 |
Cannabis |
155.431 |
7 |
0.001 |
0.053 |
0.046-0.061 |
0.013 |
0.992 |
0.982 |
0.85 |
Cocaine |
190.925 |
6 |
0.001 |
0.064 |
0.057-0.072 |
0.014 |
0.992 |
0.981 |
0.87 |
Stimulants |
161.657 |
6 |
0.001 |
0.059 |
0.051-0.067 |
0.008 |
0.995 |
0.988 |
0.92 |
Inhalants |
65.051 |
3 |
0.001 |
0.053 |
0.042-0.064 |
0.010 |
0.997 |
0.985 |
0.85 |
Sedatives |
94.670 |
7 |
0.001 |
0.041 |
0.034-0.049 |
0.010 |
0.995 |
0.990 |
0.86 |
Hallucinogens |
323.535 |
8 |
0.001 |
0.073 |
0.066-0.080 |
0.026 |
0.960 |
0.925 |
0.71 |
Opiods |
158.546 |
3 |
0.001 |
0.084 |
0.073-0.095 |
0.009 |
0.998 |
0.990 |
0.96 |
Others |
220.741 |
7 |
0.001 |
0.064 |
0.057-0.072 |
0.017 |
0.989 |
0.976 |
0.86 |
MDE |
|
|
|
|
|
|
|
|
|
Depression |
1,210.246 |
42 |
0.001 |
0.072 |
0.069-0.076 |
0.034 |
0.954 |
0.939 |
0.89 |
GA |
|
|
|
|
|
|
|
|
|
Anxiety |
74.940 |
5 |
0.001 |
0.043 |
0.035-0.052 |
0.005 |
0.998 |
0.995 |
0.93 |
PCL-5 |
|
|
|
|
|
|
|
|
|
Rexperimentation |
15.073 |
4 |
0.005 |
0.024 |
0.012-0.038 |
0.008 |
0.999 |
0.997 |
0.86 |
NACM |
229.317 |
10 |
0.001 |
0.068 |
0.061-0.076 |
0.020 |
0.983 |
0.964 |
0.86 |
Hyperarousal |
200.198 |
8 |
0.001 |
0.071 |
0.063-0.080 |
0.030 |
0.972 |
0.947 |
0.78 |
PTSD |
4535.593 |
162 |
0.001 |
0.076 |
0.074-0.078 |
0.044 |
0.916 |
0.901 |
0.93 |
Note. Test: LEC-5=Life Events Checklist, ASSIST= Alcohol, Smoking, and Substance Involvement Screening Test, MDE=Major Depressive Episode, GA= Generalized Anxiety, PCL-5= Post-traumatic Checklist. Scales: NACM= Negative Alterations in Cognition and Mood, PTSD= Post-traumatic Stress Disorder (including the two avoidance items). Categorical Variables do not show SRMR as Li (2021) recommended.“
- Figure 1 is unreadable. It should be explained more clearly what the dependent variable was. If there are six dependent variables in Figure 1, it would be clearer to rearrange six separate readable figures.
We have corrected and divided Figure 1 into two and described which variables were the dependent ones (rows 547 – 553).
Figure 1. Relative risks, with their respective 95% confidence intervals. Variables on the left column predict the variables listed on the right side of the graph (dependent variables). The upper half of the graph shows the effect of victimizing interpersonal and intimate violence on harmful AOD use. The bottom half of the graph shows the effect of AOD and victimizing interpersonal and intimate violence over perpetrating interpersonal and intimate violence.
(rows 574 – 579).
“
“
Figure 2. Relative risks, with their respective 95% confidence intervals. Variables on the left column predict variables named on the right side of the graph (dependent variables). Graph shows harmful AOD use and Victimizing Interpersonal and Intimate Violence affecting Mental Health symptoms for total sample.”
- Figure 2 would be easier to perceive in a landscape orientation.
Thanks for suggestion, Figure (now) 3 is in landscape orientation (rows 597 - 599).
Figure 3. Variables from SEM, path coefficients, and residual variances for whole sample.
- Also, not very readable is Figure 3 with the models for the four groups. Here, it was more appropriate to rearrange the results in a table.
We clarified the figure (now) 4 by growing the letters and decoloring the insignificant relationship; so, we decided to maintain the graphs to contrast the moderating effects (rows 625 - 627).
“
Figure 4. Sex and educational attainment SEMs with path coefficients, and residual variances.
- The notation of the literature is not in accordance with the rules given in the instructions to the authors. In citing literature, a comma not a semicolon should be the separator.
We have corrected the notation with commas instead of semicolons. For example (row 53):
“…(PAHO, 3)…”
